# Mixed-Curvature Decision Trees and Random Forests

**Philippe Chlenski** [1]  **Quentin Chu** [1]  **Raiyan R. Khan** [1]  **Kaizhu Du** [1]  **Antonio Khalil Moretti** [2,3]  **Itsik Pe'er** [1]

## Abstract

Decision trees (DTs) and their random forest (RF) extensions are workhorses of classification and regression in Euclidean spaces. However, algorithms for learning in non-Euclidean spaces are still limited. We extend DT and RF algorithms to product manifolds: Cartesian products of several hyperbolic, hyperspherical, or Euclidean components. Such manifolds handle heterogeneous curvature while still factorizing neatly into simpler components, making them compelling embedding spaces for complex datasets. Our novel angular reformulation respects manifold geometry while preserving the algorithmic properties that make decision trees effective. In the special cases of single-component manifolds, our method simplifies to its Euclidean or hyperbolic counterparts, or introduces hyperspherical DT algorithms, depending on the curvature. In benchmarks on a diverse suite of 57 classification, regression, and link prediction tasks, our product RFs ranked first on 29 tasks and came in the top 2 for 41. This highlights the value of product RFs as straightforward yet powerful new tools for data analysis in product manifolds. Code for our method is available at https://github.com/pchlenski/manify.

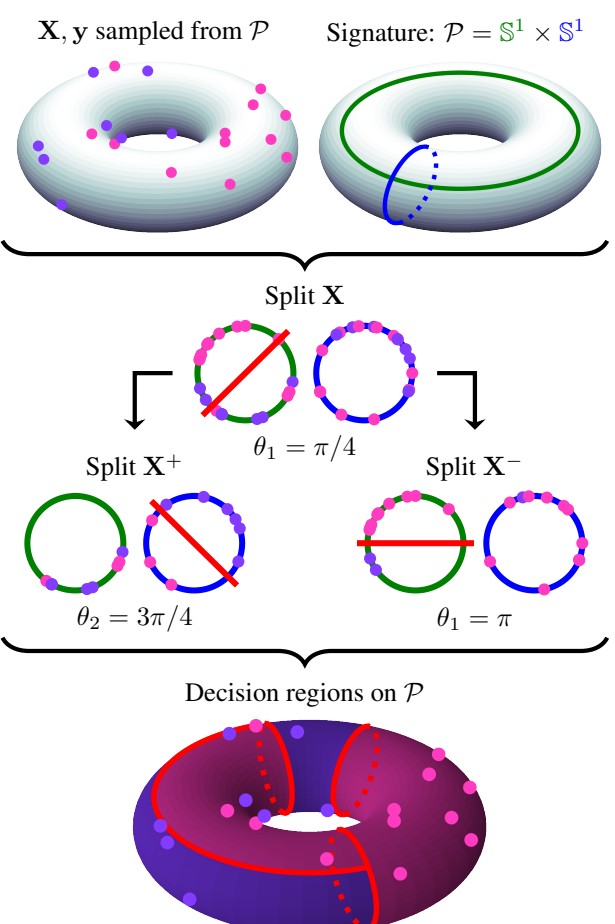

Figure 1: Given a sample of labeled points $(\mathbf{X}, \mathbf{y})$ from a torus $\mathcal{P} = \mathbb{S}^1 \times \mathbb{S}^1$, we can factorize $\mathbf{X}$ into coordinates on each component manifold. Our DT **splits** these factorized coordinates, partitioning $\mathcal{P}$ into disjoint regions (colored **positive** or **negative** to reflect the classes).

## 1. Introduction

Most machine learning algorithms assume Euclidean geometry, but real datasets often have non-Euclidean structure: tree-like hierarchies fit naturally in hyperbolic space (Sonthalia & Gilbert, 2020), while cyclical patterns suit spherical representations (Ding & Regev, 2021). Moreover, many real-world datasets don't conform to a single geometric structure. Any single constant-curvature geometry—hyperbolic, spherical, or Euclidean—struggles to capture all of their structural nuances simultaneously.

Product manifolds (Gu et al., 2018), which combine multiple constant-curvature components into a single product space, are more expressive than single manifolds, facilitating faithful representations of more kinds of underlying structure in the data. Although product manifolds have made inroads in biology (McNeela et al., 2024) and knowledge graph applications (Wang et al., 2021), general-purpose machine learning on product manifolds remains underex-

[1]Columbia University [2]Barnard College [3]Spelman College. Correspondence to: Philippe Chlenski <pac@cs.columbia.edu>.

*Proceedings of the $42^{nd}$ International Conference on Machine Learning*, Vancouver, Canada. PMLR 267, 2025. Copyright 2025 by the author(s).

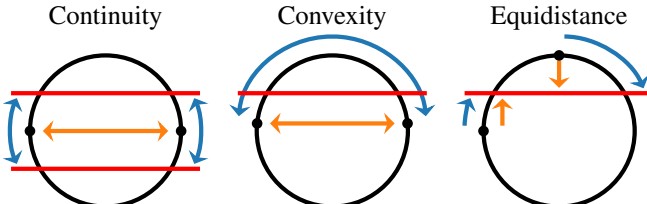

Figure 2: Decision tree **splits** are based on Euclidean distances in the **ambient space**, and their assumptions sometimes fail to hold for geodesic distances on **non-Euclidean manifolds**. For example, splits can create disconnected regions (Continuity), geodesics may cross boundaries within regions (Convexity), and distances to splits become unequal (Equidistance).

plored. In particular, decision trees (DTs) and random forests (RFs)—some of the most successful and widely-used algorithms in machine learning—still lack a product space variant.

A natural approach would be to simply apply standard DTs/RFs to the coordinate representation of points embedded in the product manifold. However, this naive strategy ignores the underlying geometry: splits that appear reasonable in coordinate space can violate fundamental geometric properties when interpreted on the manifold itself. We conjecture that the success of DTs stems from five key desiderata that standard Euclidean splits naturally preserve:

1. **Continuity**: Leaves partition space into connected regions;
2. **Convexity**: Leaves correspond to geodesically convex subsets of the input space;
3. **Equidistance**: Splits maintain equal distance to nearest points on either side;
4. **Efficiency**: Consider $\mathcal{O}(nd)$ candidates per split; and
5. **Speed**: Evaluate splits in $\mathcal{O}(1)$ time.

Figure 2 illustrates how these desiderata fail on curved manifolds: standard linear splits through the ambient space leave room for topologically discontinuous leaves, geodesics that cross decision boundaries, and unequal distances to the decision boundary. Such issues could lead to misgeneralization, undermining the geometric intuition that makes decision trees interpretable and effective.

We present a unified framework that preserves all five desiderata across arbitrary product manifolds. By representing splits as angles in two-dimensional subspaces, our approach naturally handles hierarchical data (hyperbolic components), cyclical patterns (spherical components), and traditional features (Euclidean components) within a single, principled framework.

**Our contributions:**

1. We generalize DTs and RFs to *all* constant-curvature manifolds: By representing data and splits as angles in two-dimensional subspaces, we guarantee the five desiderata above. For single manifolds, this extends existing Euclidean and hyperbolic models or introduces *hyperspherical* DTs and RFs.
2. We introduce novel DT and RF algorithms for product manifolds.
3. We extend techniques for sampling mixtures of Gaussians to product manifolds.
4. We show how problems like link prediction in graphs and signal analysis can be recast as inference problems on product manifolds.
5. We demonstrate the effectiveness of our algorithms on a diverse suite of 57 benchmarks.

### 1.1. Related work

**Non-Euclidean representation learning.** Important background on manifolds in machine learning is given in Cayton (2005) and Bengio et al. (2014). Much of the work on product manifolds is indebted to early works on hyperbolic spaces, including Nickel & Kiela (2017); Chamberlain et al. (2017), and Ganea et al. (2018).

**Machine learning in product manifolds.** Tabaghi et al. (2021) describe linear classifiers, including perceptron and support vector machines; Tabaghi et al. (2024) adapt principal component analysis; and Cho et al. (2023) generalize transformer architectures to product manifolds.

**Product manifold-derived features.** Sun et al. (2021) and Borde et al. (2023b) use product manifolds to compute rich similarity measures as features for classification. Giovanni et al. (2022) introduce a heterogeneous variant of product manifolds; Borde et al. (2024) combine quasi-metrics and partial orders for graph representation.

**Manifold random forests.** Our method is inspired by recent work by Doorenbos et al. (2023) and Chlenski et al. (2024) extending RFs to hyperbolic space. Other work has explored generalizations of random forests to manifolds, including random forest regression for manifold-valued *targets* (Tsagkrasoulis & Montana, 2017), manifold oblique random forests (Li et al., 2022), and Fréchet random forests (Capitaine et al., 2024). Chlenski & Pe'er (2025) explores faster ways to train hyperbolic random forests.

**Applications of product manifolds.** Product manifolds are used to embed knowledge graphs (Wang et al., 2021; Nguyen-Van et al., 2023; Li et al., 2024). In biology, they have been used to represent pathway graphs (McNeela et al., 2024), cryo-EM images (Zhang et al., 2021), and single-cell transcriptomic profiles (Tabaghi et al., 2021). Skopek et al. (2020) also embed image datasets into product manifolds.

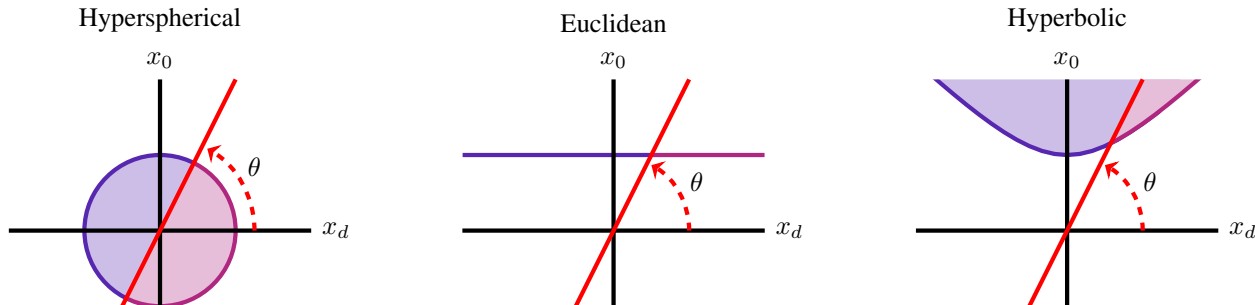

Figure 3: Decision boundaries in any constant-curvature manifold can be found by projection into 2-dimensional subspaces. Depending on the manifold, points end up inside a circle, a line, or a hyperbola; from there, data and splits can both be represented as a single angle $\theta$. Each **split** divides the manifold into **positive** and **negative** regions.

## 2. Preliminaries

We review relevant details of different Riemannian manifolds (Euclidean spaces, hyperspheres, hyperboloids, and product manifolds), along with key properties of the Euclidean and hyperbolic variants of DTs and RFs.

### 2.1. Riemannian manifolds

We will begin by reviewing key details of hyperspheres, hyperboloids, and Euclidean spaces. For more details, readers can consult Do Carmo (1992).

Each space described is a Riemannian manifold, meaning that it is locally isomorphic to Euclidean space and equipped with a distance metric. The shortest paths between two points $\mathbf{u}$ and $\mathbf{v}$ on a manifold are called geodesics. As all three spaces we consider have constant Gaussian curvature, we define simple closed forms for geodesic distances in each of the following subsections in lieu of a more general discussion of geodesics in arbitrary Riemannian manifolds.

Any constant-curvature manifold $\mathcal{M}$ is parameterized by a dimensionality $D$ and a curvature $K$, and resides in an ambient space $\mathbb{R}^{D+1}$. Finally, for each point $\mathbf{x} \in \mathcal{M}$, the tangent plane at $\mathbf{x}$, $T_{\mathbf{x}}\mathcal{M}$, is the space of all tangent vectors at $\mathbf{x}$:

$$T_{\mathbf{x}}\mathcal{M} = \{\mathbf{v} \in \mathbb{R}^{D+1} : \langle \mathbf{v}, \mathbf{x} \rangle_{\mathcal{M}} = 0\}. \tag{1}$$

#### 2.1.1. EUCLIDEAN SPACE

Euclidean spaces are naturally understood as $\mathbb{R}^D$, but we will use the notation $\mathbb{E}^D = \mathbb{R}^D$ when treating Euclidean spaces as manifolds. In contrast, we will continue to use $\mathbb{R}^D$ to refer to ambient spaces. Euclidean spaces use the familiar inner product (dot product), norm ($\ell_2$ norm), and distance function (Euclidean distance):

$$\langle \mathbf{u}, \mathbf{v} \rangle = u_0 v_0 + u_1 v_1 + \ldots + u_2 v_2, \tag{2}$$

$$\|\mathbf{u}\| = \sqrt{\langle \mathbf{u}, \mathbf{u} \rangle}, \tag{3}$$

$$\delta_{\mathbb{E}}(\mathbf{u}, \mathbf{v}) = \|\mathbf{u} - \mathbf{v}\|. \tag{4}$$

#### 2.1.2. HYPERSPHERICAL SPACE

Hyperspherical space is characterized by constant positive curvature. In such spaces, angles in any triangle sum to more than $\pi$ and there are no parallel lines. The familiar examples of circles and 2-spheres are low-dimensional examples of hyperspherical spaces.

Hyperspheres can be viewed as surfaces *embedded* in a higher-dimensional, Euclidean ambient space. Hyperspherical space uses the same inner products as Euclidean space. The hypersphere is the set of points in the ambient space having a Euclidean norm equal to some radius inversely proportional to the curvature $K > 0$:

$$\mathbb{S}^{D,K} = \{\mathbf{x} \in \mathbb{R}^{D+1} : \|x\| = 1/K\}. \tag{5}$$

Because shortest paths between two points $\mathbf{u}$ and $\mathbf{v}$ in $\mathbb{S}^{D,K}$ through the ambient space leave the surface of the manifold, we must define the hyperspherical distance function for the shortest path entirely in $\mathbb{S}^{D,K}$ between $\mathbf{u}$ and $\mathbf{v}$:

$$\delta_{\mathbb{S}}(\mathbf{u}, \mathbf{v}) = \cos^{-1}(K^2 \langle \mathbf{u}, \mathbf{v} \rangle)/K. \tag{6}$$

#### 2.1.3. HYPERBOLIC SPACE

Hyperbolic space is characterized by constant negative metric curvature. This has several consequences: for instance, the angles in any triangle sum to less than $\pi$, many lines through a point can be parallel to any given line, and neighborhoods grow exponentially with radius.

There are several equivalent models of hyperbolic space. For our purposes, we will describe the hyperbolic space from the perspective of the hyperboloid model. First, we must define the ambient Minkowski space. This is a vector space equipped with the Minkowski inner product:

$$\langle \mathbf{u}, \mathbf{v} \rangle_{\mathcal{L}} = -u_0 v_0 + u_1 v_1 + \ldots + u_n v_n. \tag{7}$$

Similar to the Euclidean case, we let $\|\mathbf{u}\|_{\mathcal{L}} = \langle \mathbf{u}, \mathbf{u} \rangle_{\mathcal{L}}$ (we do not wish to take the square root of a negative number).

The hyperboloid of dimension $D$ and curvature $K < 0$, written $\mathbb{H}^{D,K}$, is a set of points with constant Minkowski norm:

$$\mathbb{H}^{D,K} = \{\mathbf{x} \in \mathbb{R}^{D+1} \ : \ \|\mathbf{x}\|_{\mathcal{L}} = -1/K^2, \ x_0 > 0\}, \quad (8)$$

Finally, the hyperbolic distance function for geodesic distances between $\mathbf{u}, \mathbf{v} \in \mathbb{H}^{D,K}$ is given by

$$\delta_{\mathbb{H}}(\mathbf{u}, \mathbf{v}) = -\cosh^{-1}(K^2 \langle \mathbf{u}, \mathbf{v} \rangle_{\mathcal{L}})/K. \quad (9)$$

### 2.1.4. MIXED-CURVATURE PRODUCT MANIFOLDS

We reiterate the definition of product manifolds from Gu et al. (2018). Following the convention of using $\prod_i \mathbf{X_i}$ to refer to the iterated *Cartesian* product over sets, the product manifold is defined as

$$\mathcal{P} = \prod_{i=1}^{n} \mathbb{S}^{s_i, K_i} \times \prod_{j=1}^{m} \mathbb{H}^{h_j, K'_j} \times \mathbb{E}^d. \quad (10)$$

The total number of dimensions is $\sum_i^n s_i + \sum_j^m h_j + d$. Each individual manifold is called a component manifold, and the decomposition of the product manifold into component manifolds is called the signature. Informally, the signature can be considered a list of dimensionalities and curvatures for each component manifold.

Distances in $\mathcal{P}$ decompose as the $\ell_2$ norm of the distances in each of the component manifolds:

$$\delta_{\mathcal{P}}(\mathbf{u}, \mathbf{v}) = \sqrt{\sum_{\mathcal{M} \in \mathcal{P}} \delta_{\mathcal{M}}(\mathbf{u}_{\mathcal{M}}, \mathbf{v}_{\mathcal{M}})^2}, \quad (11)$$

where $\mathbf{u}_{\mathcal{M}}$ and $\mathbf{v}_{\mathcal{M}}$ denotes the restriction of $\mathbf{u}$ and $\mathbf{v}$ to their components in $\mathcal{M}$ and $\delta_{\mathcal{M}}$ refers the distance function appropriate to $\mathcal{M}$.

For $\mathbf{x} \in \mathcal{P}$, the tangent plane at $\mathbf{x}$, $T_{\mathbf{x}}\mathcal{P}$, is the direct sum (concatenation) of all component tangent planes:

$$T_{\mathbf{x}}\mathcal{P} = \bigoplus_{\mathcal{M} \in \mathcal{P}} T_{\mathbf{x}_{\mathcal{M}}} \mathcal{M}. \quad (12)$$

We additionally define the origin of $\mathcal{P}$, $\mu_0$, as the concatenation of the origins of each respective manifold. The origin is $(1/|K|, 0, \ldots)$ for $\mathbb{H}^{D,K}$ and $\mathbb{S}^{D,K}$, and $(0, 0, \ldots)$ for $\mathbb{E}^D$.

### 2.2. Decision trees and random forests

The Classification and Regression Trees (CART) (Breiman, 2017) algorithm fits a DT $\mathcal{T}$ to a set of labeled data $(\mathbf{X}, \mathbf{y})$. At each step, it greedily selects the split which partitions the dataset with maximum information gain,

$$\text{IG}(\mathbf{y}) = C(\mathbf{y}) - \frac{|\mathbf{y}^+|}{|\mathbf{y}|} C(\mathbf{y}^+) - \frac{|\mathbf{y}^-|}{|\mathbf{y}|} C(\mathbf{y}^-). \quad (13)$$

Here, $C(\cdot)$ is some impurity function (we use Gini impurity for classification and variance for regression). A splitting function $S(\cdot)$ partitions the *labels* $\mathbf{y}$ into $\mathbf{y}^+$ and $\mathbf{y}^-$ and the *input space* into decision regions. Classically, $S(\cdot)$ is a thresholding function which partitions the input space into high-dimensional boxes given dimension $d$ and threshold $\theta$:

$$S(\mathbf{x}) = \mathbb{I}\{x_d > \theta\}. \quad (14)$$

This algorithm is applied recursively to each decision region until a stopping condition is met (e.g., maximum number of splits is reached). The result is a fitted DT, $\mathcal{T}$, which can be used for inference. During inference, an unseen point $\mathbf{x}$ is passed through $\mathcal{T}$ until it reaches a leaf node corresponding to some decision region. For classification, the point is then assigned the majority label inside that region; for regression, it is assigned the mean value inside that region.

Finally, a RF is an ensemble of DTs, typically trained on a bootstrapped subsample of the points and features in $\mathbf{X}$ (Breiman, 2001).

### 2.2.1. HYPERBOLIC DECISION TREE ALGORITHMS

The hyperplane perspective on DTs is helpful background for understanding our method: mathematically, thresholding $\mathbf{x}$ on a dimension is equivalent to taking its dot product with the normal vector of a separating hyperplane $\mathbb{P}$, even in hyperbolic space. Although this is easy to compute for classical thresholding boundaries, which are zero in all dimensions but $d$, this perspective principally admits *any hyperplane* $\mathbb{P}$ as a valid decision boundary.

Naturally, considerations around choosing an appropriate (and computationally efficient) $\mathbb{P}$ abound. To this end, Chlenski et al. (2024) impose homogeneity and sparsity constraints on the hyperplanes they consider for hyperbolic DTs. In hyperbolic space, homogenous hyperplanes— hyperplanes that contain the origin *of the ambient space*— intersect $\mathbb{H}^{D,K}$ at geodesic submanifolds: that is, $\mathbb{P} \cap \mathbb{H}^{D,K}$ is closed under shortest paths *according to* $\delta_{\mathbb{H}}$. The sparsity constraint enforces that the normal vectors of $\mathbb{P}$ must be nonzero only in two positions: the timelike coordinate $x_0$ and some other $x_d$, which ensures that only $\mathcal{O}(nd)$ candidate hyperplanes are considered per split, and each decision can be computed in $\mathcal{O}(1)$ time using sparse dot products.

## 3. Mixed-curvature decision trees

For any DT, we must transform the input $\mathbf{X}$ into a set of candidate hyperplanes. To this end, we reframe and generalize the hyperplane approach of hyperbolic DTs. First, we observe that homogenous hyperplanes are geodesically convex in *any constant-curvature manifold*; therefore, we can extend the hyperbolic DT approach to $\mathbb{E}$ and $\mathbb{S}$. Second, we observe that fitting sparse, homogenous DTs is equivalent

**Algorithm 1** Product Space Decision Tree

1: **Procedure** FIT($\mathcal{P}$, **X**, **y**):
2:   **for** each component $\mathcal{M}$ of $\mathcal{P}$ **do**
3:     **for** each dimension $d > 0$ of $\mathcal{M}$ **do**
4:       $\theta_{i,d} \leftarrow \tan^{-1}(x_{i,0}/x_{i,d})$ {Eq. 15}
5:     **end for**
6:   **end for**
7:   **return** FITNODE($\boldsymbol{\Theta}$, **y**, 0)
8:
9: **Procedure** FITNODE($\boldsymbol{\Theta}$, **y**, depth):
10: **if** depth = max_depth or other stopping criteria **then**
11:   **return** Leaf(**y**)
12: **end if**
13: $(d^*, \theta^*, IG^*) \leftarrow (-, -, -\infty)$
14: **for** each dimension $d$ of $\boldsymbol{\Theta}$ **do**
15:   **for** each $\theta \in$ GETCANDIDATES($\boldsymbol{\Theta}, d$) **do**
16:     Partition $(\boldsymbol{\Theta}, \mathbf{y}) \rightarrow (\boldsymbol{\Theta}^{\pm}, \mathbf{y}^{\pm})$ via Eq. 16
17:     $IG \leftarrow$ Eq. 13 on $(\mathbf{y}^+, \mathbf{y}^-)$
18:     **if** $IG > IG^*$ **then**
19:       $(d^*, \theta^*, IG^*) \leftarrow (d, \theta, IG)$
20:     **end if**
21:   **end for**
22: **end for**
23: **if** $IG^* = -\infty$ **then**
24:   **return** Leaf(**y**)
25: **else**
26:   $\mathcal{N} \leftarrow$ Node($d^*, \theta^*$)
27:   $\mathcal{N}$.left $\leftarrow$ FITNODE($\boldsymbol{\Theta}^-, \mathbf{y}^-$, depth+1)
28:   $\mathcal{N}$.right $\leftarrow$ FITNODE($\boldsymbol{\Theta}^+, \mathbf{y}^+$, depth+1)
29:   **return** $\mathcal{N}$
30: **end if**
31:
32: **Procedure** GETCANDIDATES($\boldsymbol{\Theta}, d$):
33: $\tilde{\theta} \leftarrow$ sorted unique $\boldsymbol{\Theta}_{:,d}$
34: **return** $\{m_{\mathcal{M}}(\tilde{\theta}_i, \tilde{\theta}_{i+1})\}_i$ {Eqs. 18–22}

to thresholding on angles under 2-dimensional projections.

We consider the set of all projections onto the basis $\{x_0, x_d\}$, which are computable in $\mathcal{O}(1)$ time per projection by coordinate selection. First, we compute the projection angle:[1]

$$\theta(\mathbf{x}, d) = \tan^{-1}(x_0/x_d). \qquad (15)$$

Next, we use a modified splitting criterion to account for the geometry of angular splits:

$$S(\mathbf{x}, d, \theta) = \mathbb{I}\{\theta(\mathbf{x}, d) \in [\theta, \theta + \pi]\}. \qquad (16)$$

Once the best angle is selected, we must compute angular midpoints to select $\mathbb{P}$ that intersects $\mathcal{M}$ at a point *geodesi-*

---

[1] We use the PyTorch `arctan2` function to ensure that we can recover the full range of angles in $[0, 2\pi)$. This is essential for properly specifying decision boundaries in $\mathbb{S}$.

*cally equidistant* from the two points to either side of it (Euclidean DTs do this by sample-averaging the threshold values). Angular midpoints for each component manifold are described in the following sections and summarized in Table 5 in the Appendix.

With the angular features and manifold-informed midpoint modifications in place, the rest of the algorithm follows Section 2.2 unmodified.

### 3.1. Euclidean decision trees

To unify our treatment of all three geometries, we first reformulate the standard Euclidean DT in a geometrically-informed, albeit unconventional, way that is equivalent to the classical model.

While the intersections of homogenous hyperplanes in $\mathbb{R}^D$ with $\mathbb{E}^D$ are (trivially) convex, these lack the expressiveness of an ambient-space formulation. Thus, we embed $\mathbb{E}^D$ in $\mathbb{R}^{D+1}$ by a trivial lift:

$$\phi : \mathbb{E}^D \rightarrow \mathbb{R}^{D+1}, \ \phi(\mathbf{u}) = (1, \mathbf{u}). \qquad (17)$$

For two points $\mathbf{u}, \mathbf{v} \in \mathbb{E}^D$, the midpoint angles in $\mathbb{E}^D$ can be described in terms of the coordinates of $\mathbf{u}$ and $\mathbf{v}$ or their respective projection angles $(\theta_{\mathbf{u}}, \theta_{\mathbf{v}})$ as

$$m_{\mathbb{E}}(\mathbf{u}, \mathbf{v}) = \tan^{-1}(2/(u_d + v_d)) \qquad (18)$$
$$= \tan^{-1}(2/(\cot(\theta_{\mathbf{u}}) + \cot(\theta_{\mathbf{v}}))) \qquad (19)$$

See Appendix B for a proof that this formulation is equivalent to thresholding on basis dimensions.

### 3.2. Hyperbolic decision trees

For two points $\mathbf{u}, \mathbf{v} \in \mathbb{H}^{D,K}$, we compute $\theta_{\mathbf{u}}$ and $\theta_{\mathbf{v}}$ according to Eq 15 and follow Chlenski et al. (2024) in computing the hyperbolic midpoint angle in $\mathbb{H}^{D,K}$ as:

$$V := \frac{\sin(2\theta_{\mathbf{u}} - 2\theta_{\mathbf{v}})}{\sin(\theta_{\mathbf{u}} + \theta_{\mathbf{v}})\sin(\theta_{\mathbf{v}} - \theta_{\mathbf{u}})}, \qquad (20)$$

$$m_{\mathbb{H}}(\mathbf{u}, \mathbf{v}) = \begin{cases} \cot^{-1}(V - \sqrt{V^2 - 1}) & \text{if } \theta_{\mathbf{u}} + \theta_{\mathbf{v}} < \pi \\ \cot^{-1}(V + \sqrt{V^2 - 1}) & \text{otherwise.} \end{cases}$$
$$\qquad (21)$$

### 3.3. Hyperspherical decision trees

The hyperspherical case is quite simple, except that unlike hyperbolic space and the "lifted" Euclidean space after applying Eq 17, we lack a natural choice of $x_0$. We adopt the convention of fixing the first dimension of the embedding space as $x_0$, which intuitively corresponds to fixing a "north pole" at the origin $\mu_0 = (1/|K|, 0, \ldots)$.

Angular midpoints are particularly well-behaved in hyperspherical manifolds: given $\mathbf{u}, \mathbf{v} \in \mathbb{S}^{D,K}$, the hyperspherical

midpoint angle is computed by finding $\theta_\mathbf{u}$ and $\theta_\mathbf{v}$ using Eq 15 and taking their mean:

$$m_\mathbb{S}(\mathbf{u}, \mathbf{v}) = (\theta_\mathbf{u} + \theta_\mathbf{v})/2. \qquad (22)$$

### 3.4. Product decision tree algorithm

Intuitively, the transition from DTs in a single component manifold to a product manifold is that we now iterate over all preprocessed angles together, using the angular midpoint formula appropriate to each component. The complete pseudocode for this algorithm is given in Algorithm 1.

Letting a single DT span all components—as opposed to an ensemble of DTs, each operating in a single component— allows the model to independently allocate its splits across components according to their relevance to the task at hand. Recasting DT learning in terms of angular comparisons has three major advantages over finding planar decision boundaries directly:

1. We can consider angles under *arbitrary* linear projections (not just projections onto basis dimensions) while maintaining $\mathcal{O}(1)$ decision complexity.
2. It becomes possible to subsample the features (precomputed angles) as is typical in RFs.
3. Product manifolds can always represent additional features in a new Euclidean manifold.

### 3.5. Geodesic convexity

Having detailed our algorithm, we now demonstrate that its splits satisfy the geodesic convexity property. Establishing this is essential to our stated goals: geodesic convexity implies topological continuity. The remaining desiderata (3– 5) follow straightforwardly from our angular preprocessing and midpoint splitting strategy.

A subset of a manifold $\mathcal{S} \subseteq \mathcal{M}$ is said to be geodesically convex if $\mathbf{p}, \mathbf{q} \in \mathcal{S}$ implies that the geodesic $\gamma_{\mathbf{p},\mathbf{q}} \subseteq \mathcal{S}$. That is, for any two points in $\mathcal{S}$, all points in the shortest path between them stay in $\mathcal{S}$. Lack of geodesic convexity is a potential source of misgeneralization for models, and several classifiers in hyperbolic spaces explicitly seek to partition the feature space in a geodesically-convex manner (Cho et al., 2018; Chlenski et al., 2024).

Building on Udrişte (1994), Chapter 3.1, we present a proof sketch that if $\mathcal{S} \subset \mathcal{M}$ is geodesically convex and partitions $\mathcal{M}$ into disjoint regions $\mathcal{M}^+$ and $\mathcal{M}^-$, then these regions are geodesically convex.

By way of contradiction, suppose there exist $\mathbf{p}, \mathbf{q} \in \mathcal{M}^+$ such that their geodesic $\gamma_{\mathbf{p},\mathbf{q}}$ crosses into $\mathcal{M}^-$. Since $\mathcal{S}$ separates $\mathcal{M}^+$ and $\mathcal{M}^-$, $\gamma_{\mathbf{p},\mathbf{q}}$ must follow this path:

$$\mathcal{M}^+ \to \mathcal{S} \to \mathcal{M}^- \to \mathcal{S} \to \mathcal{M}^+. \qquad (23)$$

However, this implies the existence of $\mathbf{p}', \mathbf{q}' \in \mathcal{S}$ such that

$\gamma_{\mathbf{p}',\mathbf{q}'}$ crosses into $\mathcal{M}^-$, implying $\gamma_{\mathbf{p}',\mathbf{q}'} \nsubseteq \mathcal{S}$, contradicting our initial assumption that $\mathcal{S}$ is geodesically convex. Tabaghi et al. (2021) shows that a linear classifier (that is, a classifier inducing a geodesically convex decision boundary) with weights $\mathbf{w}$ and bias $b$ takes the form

$$\begin{aligned} l_\mathbf{w}^\mathcal{P} = \text{sign}(&\langle \mathbf{w}_\mathbb{E}, \mathbf{w}_\mathbb{E} \rangle + \alpha_\mathbb{S} \sin^{-1}(\langle \mathbf{w}_\mathbb{S}, \mathbf{x}_\mathbb{S} \rangle) \\ &+ \alpha_\mathbb{H} \sinh^{-1}(\langle \mathbf{w}_\mathbb{H}, \mathbf{x}_\mathbb{H} \rangle_\mathcal{L}) + b), \end{aligned} \qquad (24)$$

where $\mathbf{v}_\mathcal{M}$ means the restriction of some $\mathbf{v} \in \mathcal{P}$ to component manifold $\mathcal{M}$ and $\alpha_\mathcal{M}$ is a weight term.

Under our angular reformulation, this simplifies to

$$l_\mathbf{w}^\mathcal{P} = \text{sign}(x_0 \cos(\theta) - x_d \sin(\theta)), \qquad (25)$$

where $\theta$ is our splitting angle and $d$ is the dimension along which the split happens; because our split is confined to a single manifold, the weights $\alpha_\mathcal{M}$ do not affect the split. By restricting our attention to two dimensions within a single manifold, our formulation bypasses almost all of the complexity of evaluating geodesic splits in product manifolds.

This simplified form directly corresponds to our angular splitting criterion from Equation 16. Since this is an instance of the geodesically convex linear classifiers, our method is guaranteed to produce convex splits.

## 4. Benchmarks

We carried out benchmarks to evaluate which model, given a labeled set of mixed-curvature embeddings, achieves the lowest error on a test set. While we produced embeddings using a range of datasets and embedding techniques, our results focus only on performance on downstream tasks. We describe our data generation/embedding methods in more detail in Appendix Section D.

We summarize our benchmark results, with references to specific figures and tables, in Table 1.

Table 1: Benchmarks summary. The "Task" column is C (Classification), R (Regression), or LP (Link prediction). "#Top-$k$" columns count how often product DTs or RFs were among the top $k$ predictors for a given set of benchmarks.

| Manifold | Task | Ref | #Top-1 | #Top-2 | Total |
|---|---|---|---|---|---|
| Single | C | Fig. 4 | 7 | 11 | 11 |
| Single | R | Fig. 5 | 7 | 9 | 11 |
| Product | C | Tab. 2 | 6 | 10 | 18 |
| Product | R | Tab. 3 | 8 | 10 | 11 |
| Product | LP | Tab. 4 | 1 | 1 | 6 |
| **Total** | | | 29 | 41 | 57 |

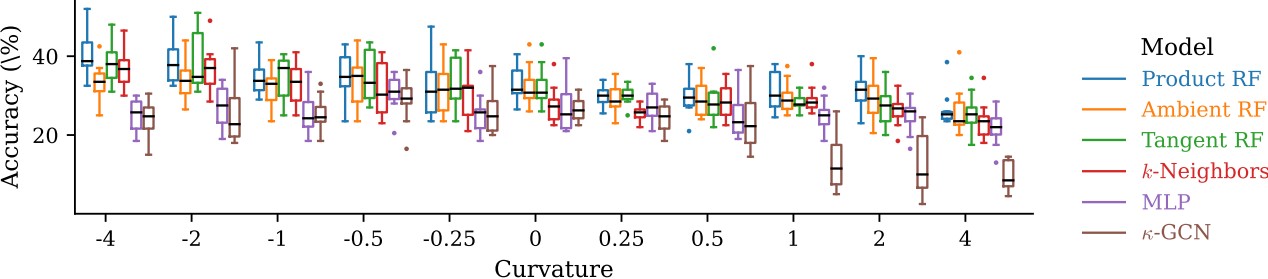

Figure 4: Classification accuracies on mixtures of 8 Gaussians in single manifolds of curvature $K$. We omit results for all models that never achieved competitive results.

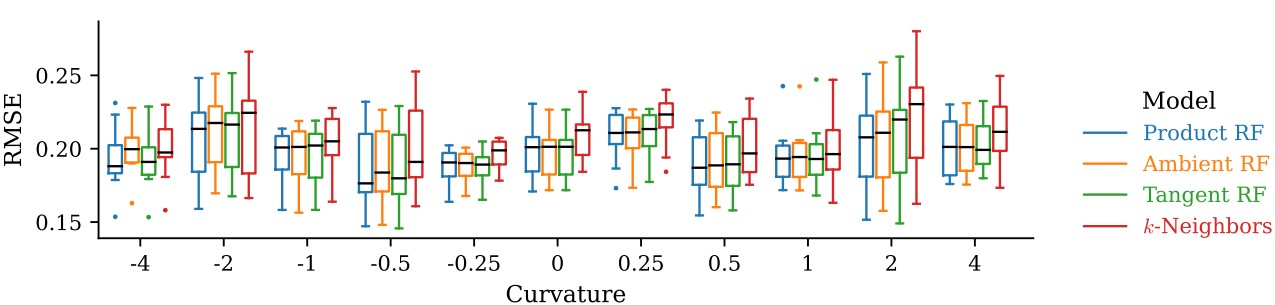

Figure 5: Regression benchmarks (RMSE) on mixtures of 8 Gaussians in single manifolds of curvature $K$. We follow the convention of Figure 4 in omitting non-competitive models.

### 4.1. Experiment details

Given a dataset $\mathbf{X}$, a set of labels $\mathbf{y}$, and a product manifold $\mathcal{P}$, we evaluate a variety of classifiers on their ability to predict $\mathbf{y}$ from $\mathbf{X}$. We apply an identical 80:20 train-test split to all of our data, train our models on the training set, and evaluate performance on the test set. We report 95% confidence intervals for accuracy scores for classification, root mean squared error (RMSE) for regression, and accuracy scores again for link prediction benchmarks. Confidence intervals are based on 10 runs with different random seeds.

### 4.2. Datasets

**Synthetic data.** We develop a novel method to sample mixtures of Gaussians in $\mathcal{P}$ to generate classification and regression datasets. For classification, we generate 8 classes using 32 clusters. For regression, we generate a single scalar response variable using 32 clusters with randomly generated intercepts. For more details, see Appendix Section A.

**Graph embeddings.** For classification and regression on graph datasets, we generate embeddings that approximate shortest-path distances in the graph using the method described in Gu et al. (2018). We select the optimal signature from the candidate set $\{(\mathbb{H}^2)^2, \mathbb{H}^2\mathbb{E}^2, \mathbb{H}^2\mathbb{S}^2, \mathbb{S}^2\mathbb{E}^2, (\mathbb{S}^2)^2, \mathbb{H}^4, \mathbb{E}^4, \mathbb{S}^4\}$ by generat-

ing embeddings in each signature and selecting the signature with the lowest metric distortion. For link prediction, we embed all datasets in $\mathcal{P} = (\mathbb{S}^2\mathbb{E}^2\mathbb{H}^2)$, then create a binary classification dataset by associating each pair of nodes with a point in $\mathcal{P}^2\mathbb{E}^1$, where each pair of points is included and the last Euclidean dimension is the manifold distance $\delta_{\mathcal{P}}(\mathbf{x_i}, \mathbf{x_j})$; labels are simply whether there is an edge between nodes $i$ and $j$. Full details on graph embeddings are described in Appendix Section D.4.

**Mixed-curvature VAE latent space.** We follow Skopek et al. (2020) in training variational autoencoders (VAEs) whose latent space is $\mathcal{P}$. Once the VAE is trained, we use its encoder to generate embeddings for our dataset and classify these embeddings. Full details on VAE training and downstream inference are described in Appendix Section D.5.

**Empirical datasets.** Some datasets can be represented in a non-Euclidean geometry without generating embeddings: for instance, geospatial data lives in $\mathbb{S}^2$, while cyclic time series embed in $\mathbb{S}^1$. We describe our approach to generating embeddings for these empirical datasets in Appendix D.6.

### 4.3. Baselines

We train two variants of Scikit-Learn (Pedregosa et al., 2011) DTs and RFs: The ambient variant operates directly on coor-

Table 2: Accuracies for all product manifold classification benchmarks. The highest scores for each dataset are shown in **bold**, while second-best predictors are underlined. For brevity, we omit columns for three low-performing methods: product space perceptrons, ambient-space GNNs, and product space MLR.

| | Dataset | Signature | Product RF | Ambient RF | Tangent RF | $k$-Neighbors | Ambient MLP | $\kappa$-GCN |
|---|---|---|---|---|---|---|---|---|
| Synthetic (multi-$K$) | Gaussian | $\mathbb{E}^4$ | **34.4 ± 3.0** | 34.2 ± 2.9 | 34.2 ± 2.9 | 34.1 ± 2.8 | 27.9 ± 3.1 | 26.5 ± 3.9 |
| | | $\mathbb{H}^4$ | **69.3 ± 3.3** | 53.1 ± 2.9 | 63.8 ± 3.4 | 67.3 ± 4.0 | 42.4 ± 5.3 | 28.8 ± 3.3 |
| | | $\mathbb{H}^2\mathbb{E}^2$ | 43.0 ± 2.8 | 40.4 ± 4.0 | 43.0 ± 3.3 | **44.7 ± 2.9** | 29.9 ± 2.7 | 26.9 ± 2.6 |
| | | $(\mathbb{H}^2)^2$ | 49.0 ± 3.0 | 42.6 ± 2.4 | 46.6 ± 2.8 | **50.8 ± 3.2** | 33.7 ± 3.9 | 26.2 ± 2.4 |
| | | $\mathbb{H}^2\mathbb{S}^2$ | **41.8 ± 2.5** | 37.3 ± 2.6 | 37.6 ± 2.7 | 38.0 ± 1.9 | 28.7 ± 2.5 | 16.4 ± 4.4 |
| | | $\mathbb{S}^4$ | 37.7 ± 2.8 | 38.4 ± 2.4 | 33.5 ± 2.2 | **40.6 ± 2.6** | 26.4 ± 1.9 | 21.1 ± 1.9 |
| | | $\mathbb{S}^2\mathbb{E}^2$ | **34.4 ± 3.0** | 33.4 ± 2.4 | 31.7 ± 2.2 | 31.6 ± 2.5 | 23.7 ± 2.0 | 16.0 ± 2.7 |
| | | $(\mathbb{S}^2)^2$ | **33.1 ± 2.5** | 32.6 ± 3.0 | 29.2 ± 2.3 | 28.1 ± 3.3 | 24.3 ± 2.1 | 15.0 ± 2.2 |
| Graph | CiteSeer | $(\mathbb{H}^2)^2$ | 26.2 ± 1.2 | **26.7 ± 1.8** | 26.3 ± 1.6 | 22.2 ± 1.7 | 23.7 ± 1.3 | 24.9 ± 1.5 |
| | Cora | $\mathbb{H}^4$ | 29.4 ± 1.1 | 29.1 ± 0.8 | 29.1 ± 0.7 | 20.8 ± 0.7 | **29.8 ± 0.9** | 29.6 ± 0.9 |
| | PolBlogs | $(\mathbb{S}^2)^2$ | 92.8 ± 0.9 | 93.0 ± 1.0 | **93.3 ± 0.5** | 92.8 ± 1.2 | 92.2 ± 0.9 | 69.9 ± 10.1 |
| VAE | Blood | $\mathbb{S}^2\mathbb{E}^2(\mathbb{H}^2)^3$ | 14.5 ± 1.4 | 18.5 ± 1.4 | **18.7 ± 1.5** | 16.9 ± 1.7 | 12.8 ± 0.9 | 11.4 ± 0.9 |
| | CIFAR-100 | $(\mathbb{H}^2)^4$ | 10.4 ± 0.9 | 10.2 ± 1.2 | **10.8 ± 1.1** | 8.4 ± 0.9 | 7.6 ± 0.9 | 5.2 ± 0.7 |
| | Lymphoma | $(\mathbb{S}^2)^2$ | **84.4 ± 2.0** | 81.4 ± 2.1 | 81.2 ± 2.2 | 78.3 ± 2.5 | 78.0 ± 0.4 | 66.9 ± 13.9 |
| | MNIST | $\mathbb{S}^2\mathbb{E}^2\mathbb{H}^2$ | 36.0 ± 5.3 | 38.1 ± 5.6 | **38.2 ± 5.5** | 34.8 ± 7.5 | 35.9 ± 6.8 | 14.0 ± 2.9 |
| Other | Landmasses | $\mathbb{S}^2$ | 88.5 ± 1.1 | 87.8 ± 1.4 | 86.2 ± 1.5 | **91.5 ± 1.1** | 72.5 ± 2.3 | 73.2 ± 2.8 |
| | Neuron 33 | $(\mathbb{S}^1)^{10}$ | 64.2 ± 2.2 | 66.3 ± 2.9 | **66.5 ± 3.0** | 50.7 ± 1.9 | 47.3 ± 2.0 | 47.3 ± 2.0 |
| | Neuron 46 | $(\mathbb{S}^1)^{10}$ | 24.1 ± 3.7 | 24.4 ± 3.5 | 22.0 ± 5.7 | 30.4 ± 4.7 | 3.1 ± 1.0 | **79.1 ± 22.8** |

Table 3: Regression performance (RMSE) for all product manifold classification benchmarks. Following the conventions of Table 2, we emphasize high-scoring predictors and omit columns for predictors that never achieve the highest score. CS PhDs is a graph embedding dataset, whereas Temperature and Traffic are empirical.

| | Dataset | Signature | Product RF | Ambient RF | Tangent RF | $k$-Neighbors |
|---|---|---|---|---|---|---|
| Synthetic (multi-$K$) | Gaussian | $\mathbb{E}^4$ | **0.023 ± 0.004** | **0.023 ± 0.003** | **0.023 ± 0.003** | **0.023 ± 0.004** |
| | | $\mathbb{H}^4$ | **0.015 ± 0.003** | 0.020 ± 0.003 | 0.017 ± 0.003 | 0.016 ± 0.003 |
| | | $\mathbb{H}^2\mathbb{E}^2$ | **0.023 ± 0.004** | 0.024 ± 0.004 | 0.024 ± 0.004 | **0.023 ± 0.004** |
| | | $(\mathbb{H}^2)^2$ | **0.021 ± 0.004** | 0.024 ± 0.003 | 0.022 ± 0.003 | **0.021 ± 0.003** |
| | | $\mathbb{H}^2\mathbb{S}^2$ | **0.025 ± 0.004** | 0.026 ± 0.004 | 0.026 ± 0.004 | 0.027 ± 0.004 |
| | | $\mathbb{S}^4$ | **0.023 ± 0.003** | **0.023 ± 0.003** | 0.024 ± 0.004 | 0.024 ± 0.004 |
| | | $\mathbb{S}^2\mathbb{E}^2$ | 0.027 ± 0.004 | 0.027 ± 0.004 | **0.027 ± 0.004** | 0.028 ± 0.005 |
| | | $(\mathbb{S}^2)^2$ | **0.028 ± 0.004** | **0.028 ± 0.005** | 0.029 ± 0.005 | 0.031 ± 0.005 |
| Other | CS PhDs | $\mathbb{H}^4$ | 205.842 ± 34.454 | 208.723 ± 36.638 | 207.872 ± 35.043 | **174.759 ± 25.074** |
| | Temperature | $\mathbb{S}^2\mathbb{S}^1$ | 77.942 ± 23.689 | **48.221 ± 23.516** | 99.561 ± 24.984 | 138.650 ± 25.130 |
| | Traffic | $\mathbb{E}(\mathbb{S}^1)^4$ | 0.307 ± 0.039 | **0.220 ± 0.017** | 0.229 ± 0.021 | 0.255 ± 0.021 |

Table 4: Accuracies for all link prediction benchmarks, also following the conventions in Table 2. The $\kappa$-GCN is trained using a conventional link prediction approach; the other classifiers are trained as binary classifiers on the product of input embeddings $\mathbf{X}^2$. All graphs were embedded into $\mathbb{H}^2\mathbb{E}^2\mathbb{S}^2$.

| Dataset | Product RF | Ambient RF | Tangent RF | $k$-Neighbors | Ambient MLP | $\kappa$-GCN |
|---|---|---|---|---|---|---|
| AdjNoun | 94.3 ± 0.0 | **94.5 ± 0.0** | 94.3 ± 0.0 | **94.5 ± 0.0** | 94.1 ± 0.0 | 94.3 ± 0.0 |
| Dolphins | **94.1 ± 0.0** | 93.5 ± 0.0 | 94.1 ± 0.0 | 90.5 ± 0.0 | **94.1 ± 0.0** | 92.9 ± 0.0 |
| Football | 91.8 ± 0.0 | 93.9 ± 0.0 | 91.8 ± 0.0 | 71.4 ± 0.0 | **95.9 ± 0.0** | **95.9 ± 0.0** |
| Karate Club | 93.9 ± 0.0 | 91.8 ± 0.0 | 87.8 ± 0.0 | 77.6 ± 0.0 | **95.9 ± 0.0** | **95.9 ± 0.0** |
| PolBooks | 91.4 ± 0.0 | 91.6 ± 0.0 | **91.8 ± 0.0** | 89.8 ± 0.0 | 91.4 ± 0.0 | 91.4 ± 0.0 |

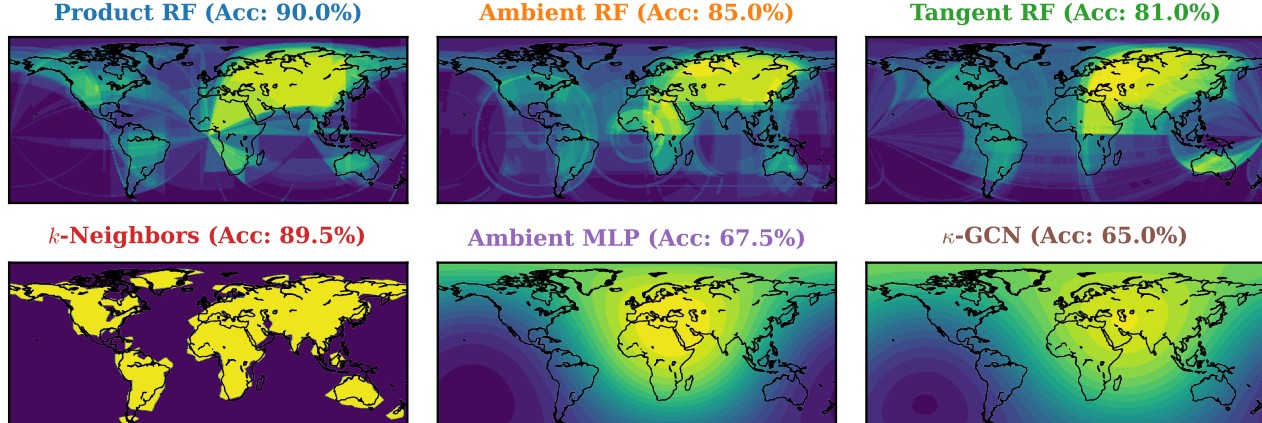

Figure 6: We color a world map with each model's predicted $\mathbb{P}(\text{Land})$ for the "Landmasses" dataset, a land vs. water classification benchmark in $\mathbb{S}^2$. Each RF consists of 12 DTs with a max depth of 5. Note the artifacts learned by **Euclidean RFs**, **tangent RFs**, and **$k$-Neighbors**, as well as the diffuse probabilities learned by the **ambient MLP** and **$\kappa$-GCN**.

dinates in the ambient space $\mathbb{R}^{D+1}$, while the tangent plane variant preprocesses points by projecting from $\mathcal{P}$ to $\mathcal{T}_{\mu_0}\mathcal{P}$ via the logarithmic map at $\mu_0$. We use $k$-nearest neighbor ($k$-NN) classifiers and regressors with precomputed pairwise distance matrices according to $\delta_{\mathcal{P}}$ (Eq. 11). We also use the product space perceptron algorithm (Tabaghi et al., 2021) and multilayer perceptron (MLP) and $\kappa$-GCN model (Bachmann et al., 2020), as implemented in the Manify library (Chlenski et al., 2025).

For our own models, we set hyperparameters identically to Scikit-Learn DTs and RFs, except we consider all $\binom{D}{2}$ projections—for a total of 3 features per 2-dimensional component manifold, just like ambient space methods use. Full details for each model can be found in Appendix D.2.

### 4.4. Results

For single-curvature synthetic datasets, our method was the best classifier in 7 out of 11 signatures (Figure 4) and the best regressor (Figure 5) for 7 out of 11 signatures. In Tables 2, 3, and 4, we demonstrate consistently good performance across a diverse range of benchmarks: We consistently outcompete baseline models in classification and regression, and perform on par for link prediction tasks. All in all, our method was the best for 29 out of 57 benchmarks (51%), and was in the top-2 best for 41 out of 57 benchmarks (72%).

Further experiments can be found in the Appendix: we provide ablations in Appendix E, detailed tables and latent space visualizations in Appendix F, runtime and computational complexity analysis in Appendix G, interpretability experiments in Appendix H, and benchmarks on more manifolds and baselines in Appendix I.

### 5. Conclusion

We present strong evidence supporting the use of mixed-curvature DTs and RFs. In particular, we motivate and describe our entire algorithm and demonstrate its effectiveness across a highly diverse set of 57 benchmarks.

Product DTs and RFs offer a valuable balance of expressiveness and simplicity, positioned between extremely legible but underpowered linear classifiers and powerful but uninterpretable neural networks operating in product manifolds. We believe that these qualities, combined with their demonstrated performance across our benchmark datasets, are compelling evidence of our method's usefulness in a non-Euclidean data analysis toolkit.

**Limitations.** While our work is downstream of signature selection and embedding generation, it relies heavily on the availability of good product manifold embeddings. Product manifolds face challenges when selecting appropriate signatures (Borde et al., 2023a), and representing certain complex patterns in data (Borde & Kratsios, 2023). Furthermore, generating embeddings can be computationally intensive. Finally, the lack of a privileged basis (Elhage et al., 2023) in non-Euclidean embeddings makes the inductive bias of decision trees less well-motivated compared to the classical tabular setting.

**Future work.** It may be possible to exploit non-privileged basis dimensions using approaches such as rotation forests (Bagnall et al., 2020), random 2-D subspace angles, or oblique decision trees. A continuous unification of all three geometries, like the $\kappa$-stereographic model described in (Skopek et al., 2020), may be more robust and elegant. Extensions to simplex geometry (Mishra et al., 2020) are also worth considering.

## Acknowledgements

This work was funded by NSF GRFP grant DGE-2036197.

## Impact statement

This paper presents work whose goal is to advance the field of Machine Learning. There are many potential societal consequences of our work, none of which we feel must be specifically highlighted here.

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

# A. Gaussian mixture details

## A.1. Overall structure

The structure of our sampling algorithm is as follows. Note that, rather than letting $\mathcal{M}$ be a manifold of arbitrary curvature, we force its curvature to be one of $\{-1, 0, 1\}$ for implementation reasons. This necessitates rescaling steps, which take place in Equations 32, 36, and 42. The result is equivalent to performing the equivalent steps, without rescaling, on a manifold of the proper curvature.

1. Generate $\mathbf{c}$, a vector that divides $m$ samples into $n$ clusters:

$$\mathbf{p_{raw}} = \langle p_0, p_1, \ldots, p_{n-1} \rangle \tag{26}$$
$$p_i \sim \text{Uniform}(0, 1) \tag{27}$$
$$\mathbf{p_{norm}} = \frac{\mathbf{p_{raw}}}{\sum_{i=0}^{n-1} p_i} \tag{28}$$
$$\mathbf{c} = \langle c_0, c_1, \ldots c_{m-1} \rangle \tag{29}$$
$$c_i \sim \text{Categorical}(n, \mathbf{p_{norm}}) \tag{30}$$

2. Sample $\mathbf{M_{euc}}$, an $n \times D$ matrix of $n$ class means:

$$\mathbf{M_{euc}} = \langle \mathbf{m_0}, \mathbf{m_1}, \ldots, \mathbf{m_{n-1}} \rangle^T \tag{31}$$
$$\mathbf{m_i} \sim \mathcal{N}(0, \sqrt{K}\mathbf{I}). \tag{32}$$

3. Move $\mathbf{M_{euc}}$ into $T_0\mathcal{M}$, the tangent plane at the origin of $\mathcal{M}$, by applying $\psi : \mathbf{x} \to (0, \mathbf{x})$ per-row to $\mathbf{M_{euc}}$:

$$\mathbf{M_{tan}} = \langle \psi(\mathbf{m_0}), \psi(\mathbf{m_1}), \ldots \psi(\mathbf{m_{n-1}}) \rangle^T, \tag{33}$$
$$\psi : \mathbb{R}^D \to \mathbb{R}^{D+1}, \ \mathbf{x} \to \langle 0, \mathbf{x} \rangle. \tag{34}$$

4. Project $\mathbf{M_{tan}}$ onto $\mathcal{M}$ using the exponential map from $T_0\mathcal{M}$ to $\mathbf{M_{tan}}$:

$$\mathbf{M} = \exp_0(\mathbf{M_{tan}}). \tag{35}$$

5. For $0 \leq i < n$, sample a corresponding covariance matrix. Here, $\sigma$ is a variance scale parameter that can be set:

$$\mathbf{\Sigma_i} \sim \text{Wishart}(\sigma\sqrt{K}\mathbf{I}, D) \tag{36}$$

6. For $0 \leq j < m$, sample $\mathbf{X_{euc}}$, a matrix of $m$ points according to their clusters' covariance matrices:

$$\mathbf{X_{euc}} = \langle \mathbf{x_0}, \mathbf{x_1}, \ldots \mathbf{x_{m-1}} \rangle^T \tag{37}$$
$$x_j \sim \mathcal{N}(0, \mathbf{\Sigma_{c_j}}). \tag{38}$$

7. Apply $\psi(\cdot)$ from Eq 34 to each $\mathbf{x_j}$ to move it into $T_0\mathcal{M}$:

$$\mathbf{X_{tan}} = \langle \psi(\mathbf{x_0}), \psi(\mathbf{x_1}), \ldots \psi(\mathbf{x_{m-1}}) \rangle^T. \tag{39}$$

8. For each row in $\mathbf{X_{tan}}$, apply parallel transport from $T_0\mathcal{M}$ to its class mean:

$$\mathbf{X_{PT}} = \langle \mathbf{x_{0,\mu}}, \mathbf{x_{1,\mu}}, \ldots, \mathbf{x_{m-1,\mu}} \rangle \tag{40}$$
$$\mathbf{x_{j,\mu}} = PT_{0 \to \mathbf{m_{c_j}}}(\mathbf{x_j}) \tag{41}$$

9. Use the exponential map at $T_\mu\mathcal{M}$ to move the points onto the manifold:

$$\mathbf{X_\mathcal{M}} = \langle \mathbf{x_{0,\mathcal{M}}}, \mathbf{x_{1,\mathcal{M}}}, \ldots, \mathbf{x_{m-1,\mathcal{M}}} \rangle \tag{42}$$
$$\mathbf{x_{j,\mathcal{M}}} = \frac{\exp_{\mathbf{m_{c_j}}}(\mathbf{x_{j,\mu}})}{\sqrt{K}} \tag{43}$$

10. Repeat steps 2–9 for as many manifolds as desired; produce a final embedding by concatenating all component embeddings column-wise:

$$\mathbf{X} = \langle \mathbf{X_{\mathcal{M}_0}}, \mathbf{X_{\mathcal{M}_1}}, \ldots \mathbf{X_{\mathcal{M}_p}} \rangle \tag{44}$$

## A.2. Equations for manifold operations

First, we provide the forms of the parallel transport operation in hyperbolic, hyperspherical, and Euclidean spaces:

$$\mathrm{PT}^{\mathbb{H}}_{\nu \to \mu}(\mathbf{v}) = \mathbf{v} + \frac{\langle \mu - \alpha\nu, \nu \rangle_{\mathcal{L}}}{\alpha + 1}(\nu + \mu) \tag{45}$$

$$\alpha = -\langle \nu, \mu \rangle_{\mathcal{L}} \tag{46}$$

$$\mathrm{PT}^{\mathbb{S}}_{\nu \to \mu}(\mathbf{v}) = \mathbf{v}\cos(d) + \frac{\sin(d)}{d}(\mu - \cos(d)\nu) \tag{47}$$

$$d = \cos^{-1}(\nu \cdot \mu) \tag{48}$$

$$\mathrm{PT}^{\mathbb{E}}_{\nu \to \mu}(\mathbf{v}) = \mathbf{v} + \mu - \nu. \tag{49}$$

The exponential map is defined as follows in each of the three spaces:

$$\exp^{\mathbb{H}}_{\mu}(\mathbf{u}) = \cosh(\|\mathbf{u}\|_{\mathcal{L}})\mu + \sinh(\|\mathbf{u}\|_{\mathcal{L}})\frac{\mathbf{u}}{\|\mathbf{u}\|_{\mathcal{L}}} \tag{50}$$

$$\exp^{\mathbb{S}}_{\mu}(\mathbf{u}) = \cos(\|\mathbf{u}\|)\mu + \sin(\|\mathbf{u}\|)\frac{\mathbf{u}}{\|\mathbf{u}\|} \tag{51}$$

$$\exp^{\mathbb{E}}_{\mu}(\mathbf{u}) = \mathbf{u}. \tag{52}$$

## A.3. Generating classification targets

To generate classification targets covering $p \leq n$ classes, all we need to do is map clusters to classes. To ensure that each class has at least one associated cluster, we arbitrarily assign the first $p$ clusters to the first $p$ classes. In the $p = n$ case, this is equal to the $p$-dimensional identity matrix, and we conclude. In the $p < n$ case, we assign the remaining $n - p$ by drawing assignments from a uniform categorical distribution over the $p$ classes.

## A.4. Generating regression targets

To generate regression targets, we draw per-cluster slopes and intercepts:

$$\beta_{i,k} \sim Uniform(-1, 1) \tag{53}$$
$$\alpha_i \sim Uniform(-10, 10 \tag{54}$$

We then multiply each $x_j \in \mathbf{X_e}\mathbf{uc}$ (i.e. the pre-transport samples from the normal distribution) by $\beta$ and add $\alpha$:

$$y_j = \mathbf{x}_j\beta + \alpha + \varepsilon \tag{55}$$

To make the regression task more constrained and, therefore, to make the RMSEs across samples more comparable, we further normalize the labels to the range $[0, 1]$ by subtracting the minimum $y$ value and dividing by the range.

## A.5. Relationship to other work

Nagano et al. (2019) developed the overall technique used for a single cluster and a single manifold, i.e. steps 6–9. Chlenski et al. (2024) modified this method to work for mixtures of Gaussians in $\mathbb{H}^{d,1}$, and deployed it for $d \in 2, 4, 8, 16$. This corresponds to steps 1–5 of our procedure (although note that our covariance matrices are sampled differently in step 5). Thus, our contribution is simply to add step 10, modify step 5 to use the Wishart distribution, to add curvature-related scaling factors in Equations 32, 36, and 42, and to generate classification and regression targets as described in the preceding sections.

We apply this to *hyperspherical* manifolds, for which the von Mises-Fisher (VMF) distribution is typically preferred. This is an unconventional choice, but has been employed previously by Skopek et al. (2020) in their mixed-curvature VAE formulation. We do not argue for the superiority of our approach over the VMF distribution in general; however, we prefer to use ours for these benchmarks, as it allows us to draw simpler parallels between manifolds of different curvatures.

# B. Proof of equivalence for Euclidean case

A classical CART tree splits data points according to whether their value in a given dimension is greater than or less than some threshold value $t$. Midpoints are simple arithmetic means. This can be written as:

$$S'(\mathbf{x}, d, t) = \begin{cases} 1 \text{ if } x_d > t, \\ 0 \text{ otherwise.} \end{cases} \tag{56}$$

$$m_{DT}(\mathbf{u}, \mathbf{v}) = \frac{u_d + v_d}{2}. \tag{57}$$

In our transformed DT, we lift the data points by applying $\phi : \mathbf{x} \to (1, \mathbf{x})$ and then check which side of an axis-inclined hyperplane they fall on. The splitting function is based on the angle $\theta$ of inclination with respect to the $(0, d)$ plane, i.e., $\langle 1, x_d \rangle$. Our midpoints are computed to ensure equidistance in the original manifold:

$$S(\mathbf{x}, d, \theta) = \text{sign}(\sin(\theta)x_d - \cos(\theta)x_0) \tag{58}$$

$$m_{\mathbb{E}}(\mathbf{u}, \mathbf{v}) = \tan^{-1}\left(\frac{2}{u_d + v_d}\right) \tag{59}$$

To demonstrate the equivalence of the classical DT formulation to our transformed algorithm in $\mathbb{E}$, we will show that Equation 56 is equivalent to Equation 58 and Equation 57 is equivalent to Equation 59 under

$$\theta = \cot^{-1}(t). \tag{60}$$

## B.1. Equivalence of Splits

First, we show that Equations 56 and 58 are equivalent, assuming $t \neq 0$:

$$S(\mathbf{x}, d, \theta) = \text{sign}(\sin(\theta)x_d - \cos(\theta)x_0) = 1 \tag{61}$$

$$\iff \sin(\theta)x_d - \cos(\theta) > 0 \tag{62}$$

$$\iff \frac{\sin(\theta)}{\cos(\theta)}x_d = \tan(\theta)x_d > 1 \tag{63}$$

$$\iff x_d/t > 1 \tag{64}$$

$$\iff x_d > t \tag{65}$$

$$\iff S'(\mathbf{x}, d, t) = 1 \tag{66}$$

## B.2. Equivalence of midpoints

Now, we show that Equations 57 and 59 are equivalent:

$$\cot^{-1}(m_{DT}(\mathbf{u}, \mathbf{v})) = \cot^{-1}\left(\frac{u_d + v_d}{2}\right) \tag{67}$$

$$= \tan^{-1}\left(\frac{2}{u_d - v_d}\right) \tag{68}$$

$$= m_{\mathbb{E}}(\mathbf{u}, \mathbf{v}) \tag{69}$$

# C. Summary of angular midpoint formulas

Table 5: Distance functions and midpoint angle formulas for each component manifold type.

| Manifold $\mathcal{M}$ | Distance $\delta_{\mathcal{M}}(\mathbf{u}, \mathbf{v})$ | Midpoint angle $\theta_{\mathcal{M}}(\mathbf{u}, \mathbf{v})$ |
|---|---|---|
| $\mathbb{S}^{D,K}$ | $\cos^{-1}\left(\dfrac{K^2\langle\mathbf{u},\mathbf{v}\rangle}{K}\right)$ | $\dfrac{\theta_{\mathbf{u}} + \theta_{\mathbf{v}}}{2}$ |
| $\mathbb{E}^{D,0}$ | $\sqrt{\langle\mathbf{u},\mathbf{v}\rangle}$ | $\tan^{-1}\left(\dfrac{2}{u_d + v_d}\right)$ |
| $\mathbb{H}^{D,K}$ | $\dfrac{-\cosh^{-1}(K^2\langle\mathbf{u},\mathbf{v}\rangle_{\mathcal{L}})}{K}$ | $\cot^{-1}(V - \sqrt{V^2-1})$ if $\theta_{\mathbf{u}} + \theta_{\mathbf{v}} < \pi$, $\cot^{-1}(V + \sqrt{V^2-1})$ otherwise. $V := \dfrac{\sin(2\theta_{\mathbf{u}} - 2\theta_{\mathbf{v}})}{\sin(\theta_{\mathbf{u}} + \theta_{\mathbf{v}})\sin(\theta_{\mathbf{v}} - \theta_{\mathbf{u}})}$ |

# D. Full benchmark details

## D.1. Product DT/RF hyperparameters

For our models, we set the `n_features = "n_choose_2"` parameter. This means that we consider all $\binom{n}{2}$ linear projections. We do this because we restrict ourselves to 2-dimensional component manifolds, and therefore we only observe $\binom{3}{2} = 3$ total angles, equal to the number of features used by ambient space Euclidean methods. All other hyperparameters are set identically to the scikit-learn DT/RF settings below.

## D.2. Scikit-learn hyperparameters

**Random forests and decision trees.** For fairness, we set all DT and RF hyperparameters identically. Specifically, we set the following hyperparameters for both DTs and RFs:

- `max_depth = 5`
- `min_samples_split = 2`
- `min_samples_leaf = 1`
- `min_impurity_decrease = 0.0`

For RFs, we also set the following hyperparameters:

- `n_estimators = 12`
- `max_features = "sqrt"`
- `bootstrap = True` (subsamples the training data)
- `max_samples = None` (draws $n$ samples from a set of $n$ points)

Because the scikit-learn implementation differs substantially from ours, subsamples vary even when the random seed is set. Nevertheless, we also employ the same random seed for all RF models.

$k$-**nearest neighbor models.** For $k$-nearest neighbors, we use default hyperparameters.

**Product space perceptrons and SVMs.** Product space perceptrons only have one hyperparameter, which is the relative weight assigned to each component manifold. We elect to give each component manifold equal weight.

Neither the SVM code provided by Tabaghi et al. (2021) nor our own reimplementation would run on our datasets. In particular, we had issues satisfying the convexity constraints described in their paper, causing the solver to crash. Correcting this mistake and augmenting our benchmarks with SVM evaluations is a direction for future research.

### D.3. Neural networks

**$\kappa$-GCN overview.** Neural networks, especially graph neural networks, are a popular choice for representing and working with mixed-curvature representations (Sun et al., 2021; Cho et al., 2023; Bachmann et al., 2020; McNeela et al., 2024). In particular, we use the Manify (Chlenski et al., 2025) implementation of $\kappa$-GCNs, described in Bachmann et al. (2020), as the basis for our neural network models. This model is heavily inspired by prior work on generalizing graph convolutional networks (GCNs) in hyperbolic spaces (Chami et al., 2019; Ganea et al., 2018). Since the $\kappa$-GCN model uses different models of non-Euclidean space (the Poincaré disk for hyperbolic space and the projected sphere for hyperspherical space), we transform our points to these spaces using the standard stereographic projection:

$$\phi(\langle x_0, x_1, \ldots, x_D \rangle) \to \left\langle \frac{x_1}{1 + \sqrt{|K|}x_0}, \frac{x_1}{1 + \sqrt{|K|}x_0}, \ldots, \frac{x_D}{1 + \sqrt{|K|}x_0} \right\rangle. \tag{70}$$

At each layer, the $\kappa$-GCN applies a weight matrix $\mathbf{W}$ and aggregates updated embeddings using an adjacency matrix $\mathbf{A}$ (analogous to the traditional GCN update operation, $\mathbf{H}^{(l+1)} = \sigma(\mathbf{A}\mathbf{H}^l\mathbf{W})$, except using manifold-appropriate variants of left- and right-matrix multiplication, and applying the nonlinearity through logarithmic and exponential maps. At the final layer, the $\kappa$-GCN computes stereographic logits:

$$\mathbb{P}(y = k \mid \mathbf{x}) = \text{Softmax}\left(\text{logits}_{\mathcal{M}}(\mathbf{x}, k)\right) \tag{71}$$

$$\text{logits}_{\mathcal{M}}(\mathbf{x}, k) = \frac{\|\mathbf{a_k}\|_{\mathbf{p_k}}}{\sqrt{K}} \sin_K^{-1}\left(\frac{2\sqrt{|K|}\langle \mathbf{z_k}, \mathbf{a_k}\rangle}{(1 + K\|\mathbf{z_k}\|^2)\|\mathbf{a_k}\|}\right), \tag{72}$$

where $\mathbf{a_k}$ is a column vector of the final weight matrix corresponding to class $k$, $\mathbf{p_k} \in \mathcal{M}$ is a bias term, and $\mathbf{z_k} = -\mathbf{p_k} \oplus \mathbf{x}$, with $\oplus$ denoting the Möbius addition operation. It is not totally apparent in Bachmann et al. (2020) how the logits aggregate across different product manifolds; we follow Cho et al. (2023) in aggregating logits as the the $\ell_2$-norm of component manifold logits, multiplied by the sign of the sum of the component inner products:

$$\text{logits}_{\mathcal{P}}(\mathbf{x}, k) = \sqrt{\sum_{\mathcal{M} \in \mathcal{P}} \text{logits}_{\mathcal{M}}(\mathbf{x}_{\mathcal{M}}, k)} \cdot \sum_{\mathcal{M} \in \mathcal{P}} \langle \mathbf{x}_{\mathcal{M}}, \mathbf{a_k}_{\mathcal{M}}\rangle \tag{73}$$

Intuitively, this generalizes the notion that output logits correspond to signed distances from some hyperplane specified by the column vectors and biases of the final layer; all modifications to the standard logit formula simply reflect the behavior of distances in these manifolds.

**Implementation and variants.** We use our own implementation of the $\kappa$-GCN architecture, loosely based on the implementation of stereographic logits given in Cho et al. (2023). The $\kappa$-GCN class can be manipulated in several ways: in the $K = 0$ case, it behaves exactly like a Euclidean graph convolutional network; when $\mathbf{A} = \mathbf{I}$, i.e. the adjacency matrix provided is trivial, it behaves like a manifold-appropriate version of an MLP. We use this to derive the neural models we benchmark as follows:

Table 6: A summary of the neural models benchmarked in our work. Here, $\mathcal{M}_{\text{stereo}}$ denotes the stereographic projection of $\mathcal{M}$, $D$ means the ambient dimension of $\mathcal{M}$, and $\phi(\cdot)$ is the stereographic projection.

| Model | Preprocessing | Manifold | Hidden dimensions | $\mathbf{A}$ |
|---|---|---|---|---|
| Ambient MLP | — | $\mathbb{E}^D$ | $(\mathbb{E}^{32}, \mathbb{E}^{32})$ | $\mathbf{I}$ |
| Tangent MLP | $\log_{\mu_0}(\mathbf{X})$ | $\mathbb{E}^D$ | $(\mathbb{E}^{32}, \mathbb{E}^{32})$ | $\mathbf{I}$ |
| Ambient GCN | — | $\mathbb{E}^D$ | $(\mathbb{E}^{32}, \mathbb{E}^{32})$ | $\mathbf{A}$ |
| Tangent GCN | $\log_{\mu_0}(\mathbf{X})$ | $\mathbb{E}^D$ | $(\mathbb{E}^{32}, \mathbb{E}^{32})$ | $\mathbf{A}$ |
| $\kappa$-GCN | $\phi(\mathbf{X})$ | $\mathcal{M}_{\text{stereo}}$ | $(\mathcal{M}_{\text{stereo}})$ | $\mathbf{A}$ |
| $\kappa$-MLR | $\phi(\mathbf{X})$ | $\mathcal{M}_{\text{stereo}}$ | $()$ | $\mathbf{I}$ |

**Generating adjacency matrices.** For $\kappa$-GCN to work correctly, it is important that the adjacency matrix be correctly normalized. We use a standard method, as described in Bachmann et al. (2020), to generate appropriate adjacency matrices.

For some adjacency matrix $\mathbf{A}$, we do:

$$\mathbf{A}' = \mathbf{A} + \mathbf{A}^T \tag{74}$$

$$\tilde{\mathbf{A}} = \mathbf{A}' + \mathbf{I} \tag{75}$$

$$\tilde{D}_{ii} = \sum_k \tilde{A}_{ik} \tag{76}$$

$$\hat{\mathbf{A}} = \tilde{\mathbf{D}}^{-1/2} \tilde{\mathbf{A}} \tilde{\mathbf{D}}^{-1/2}, \tag{77}$$

i.e. $\mathbf{A}'$ is $\mathbf{A}$ made symmetric, $\tilde{\mathbf{A}}$ has self-connections, $\mathbf{D}$ is a diagonal matrix of row-wise degree sums, and $\hat{\mathbf{A}}$ is the properly normalized version of $\mathbf{A}$.

When an adjacency matrix is not provided (i.e., for all benchmarks except the graph embeddings), we compute $\mathbf{A}$ via a standard Gaussian kernel on the normalized pairwise distances between points:

$$A_{i,j} = \exp\left( \frac{-\delta_{\mathcal{M}}(\mathbf{x_i}, \mathbf{x_j})}{\max_{k,l} \delta(\mathbf{x_k}, \mathbf{x_l})} \right), \tag{78}$$

followed by the transformation from $\mathbf{A}$ to $\hat{\mathbf{A}}$ as described above.

**Classification.** The stereographic logits described above can be turned into classification targets through a standard softmax function.

**Regression.** For regression problems, we set the output dimension of our $\kappa$-GCN to 1 and skip the final softmax. We use a mean squared error loss function to train. This variant of $\kappa$-GCN recapitulates the typical relationship between regression and classification; however, it has not previously been described. In our experience, it unfortunately tends to grossly underperform other models.

**Link prediction.** The link prediction variant of the $\kappa$-GCN was also not described in Bachmann et al. (2020). We follow a closely related paper, Chami et al. (2019), in the standard choice of applying the Fermi-Dirac decoder (Krioukov et al., 2010; Nickel & Kiela, 2017) to predict edges:

$$\mathbb{P}((i,j) \in \mathcal{E} | \mathbf{x_i}, \mathbf{x_j}) = \left( \exp\left( \frac{\delta_{\mathcal{M}}(\mathbf{x_i}, \mathbf{x_j})^2 - r}{t} \right) + 1 \right)^{-1}, \tag{79}$$

where the embeddings for points $i$ and $j$, $\mathbf{x_i}$ and $\mathbf{x_j}$, may be updated by $\kappa$-GCN layers.

**Shared hyperparameters.** For all neural networks, we used a learning rate of .0001 and trained for 4,000 epochs. For Euclidean parameters, we used Adam (Kingma & Ba, 2017), whereas for non-Euclidean parameters we used Riemannian Adam (Becigneul & Ganea, 2018) implemented in Geoopt (Kochurov et al., 2020). Both optimizers use the hyperparameters $\beta_1 = 0.9$ and $\beta_2 = 0.999$. These hyperparameters were chosen on the basis of their convergence and good performance in exploratory hyperparameter sweeps.

### D.4. Graph embeddings

**Learning embeddings.** We reimplement the method in Gu et al. (2018) to learn graph embeddings. In particular, we use the NetworkX package (Hagberg et al., 2008) to load the graph, extract the largest connected component, and compute pairwise distances between nodes using the Floyd-Warshall algorithm. For embedding purposes, we treat all graphs as undirected. Pairwise distances were normalized into the range $[0, 1]$ by dividing by the maximum distance. To prevent train-test leakage, we take a non-transductive learning approach and mask out the gradients from the test nodes to the training nodes during the embedding process.

**Embedding hyperparameters.** Embeddings were learned using Riemannian Adam (Becigneul & Ganea, 2018) implemented in Geoopt (Kochurov et al., 2020). For each signature, we train 10 randomly-initialized embeddings for 10,000 epochs each. We treat the first 2,000 epochs as a burn-in period, during which the learning rate is .001 and the curvature of each manifold is fixed. For the remaining epochs, we train embedding coordinates with a learning rate of 0.01 and scale factors with a learning rate of 0.001. These hyperparameters were chosen based on their stability and convergence in exploratory experiments.

**Train-test split.** We avoid train-test leakage during embeddings generation by masking the gradients from the test set to the training set. Similarly, we performed the train-test split at the node level for all tasks including link prediction, meaning there was not leakage through the adjacency matrix.

**Evaluations.** Since it was not clear *a priori* which signature would embed each graph the best, we learned 10 embeddings for each candidate signature and took the one with the best $D_{avg}$ to be the benchmark signature. Our reasoning is that the lowest-distortion embedding of the graph is the most appropriate benchmark for evaluating the geometrical appropriateness of a classifier.

**Link prediction.** To generate link prediction datasets, we trained 100 randomly initialized sets of node embeddings in $\mathbb{S}^2 \times \mathbb{E}^2 \times \mathbb{H}^2$. If we let $\mathbf{X}$ be our original node embeddings and $\mathcal{E}$ be the ground-truth edges of the graph, we then generated the following dataset:

$$\mathbf{X}_{LP} = \{\langle \mathbf{x}_i, \mathbf{x}_j \rangle \text{ for } (\mathbf{x}_i, \mathbf{x}_j, \delta_{\mathcal{P}}(\mathbf{x}_i, \mathbf{x}_j) \in \mathbf{X}\} \tag{80}$$

$$\mathbf{y}_{LP} = \{\mathbb{I}\{(\mathbf{x}_i, \mathbf{x}_j) \in \mathcal{E}\} \text{ for } (\mathbf{x}_i, \mathbf{x}_j) \in \mathbf{X}\} \tag{81}$$

The corresponding signature is $(\mathcal{P})^2 \times \mathbb{E}^1$; in the case of our embeddings, that is $(\mathbb{S}^2 \times \mathbb{E}^2 \times \mathbb{H}^2)^2 \times \mathbb{E}^1$. For GCN-based models which use an adjacency matrix, we applied a Gaussian kernel to the normalized pairwise distances as described in Equation 78; this prevents the labels from leaking into the training process through the adjacency matrix.

### D.5. VAE training

**Encoder/decoder architectures.** Following Tabaghi et al. (2021), we use the following encoder/decoder architectures:

- Lymphoma dataset: Two 200-dimensional hidden layers, 500 epochs
- Blood cell dataset: Three 400-dimensional hidden layers, 200 epochs
- Omniglot and MNIST: 400-dimensional latent
- CIFAR-100: $4 \times 4$ convolutional kernels with stride 2 and padding 1. Encoder: 3 CNN layers of 64, 128, and 512 channels. Decoder: 2048-dimensional dense layer, followed by 2 CNN layers of 256, 64, and 3 channels.

**Training hyperparameters.** Our VAEs were trained using the Adam optimizer (Kingma & Ba, 2017) with default parameters (learning rate .001, $\beta_1 = 0.9$, $\beta_2 = 0.999$. In all models, each layer except the last is followed by a ReLU activation function. Curvatures were trained identically, except using a learning rate of .0001, after 100 burn-in epochs. Because some training details were omitted from the original papers, we additionally chose the following hyperparameters:

- Batch size: 4,096
- Number of samples per point: 64
- $\beta$ (weight for KL-divergence in VAE loss): 1

**Train-test split.** To minimize the risk of data leakage, we trained our VAEs on only the training data, then used the trained VAEs to generate embeddings for the training and test data. Embeddings were generated by running points through the VAE encoder and taking the returned mean parameter.

**Evaluations.** To conserve memory, we randomly subsampled 1,000 points from the training and test sets for each evaluation. We ran 10 trials per dataset in total.

### D.6. Empirical datasets

**Landmasses.** We generated a geospatial classification dataset for land versus water prediction by sampling 1,000 points from an evenly sampled grid of 10,000 longitudes and latitudes, transforming them to 3-dimensional coordinates, and assigning a "land" or "water" label to each point using the Basemap library in Matplotlib (Hunter, 2007). For classification, we associate the 3-dimensional coordinates with the signature $\mathbb{S}^2$.

**Neural spiking prediction.** We use patch-clamp electrophysiology datasets downloaded from the Allen Mouse Brain Atlas (Jones et al., 2009). We arbitrarily pick Neurons 33 and 46 for their nontrivial spiking dynamics. We perform a train-test split by taking the first 80% of time points for training, and holding out the rest for testing. To represent signals in product spaces, we apply a Fast Fourier Transform to the training data and take the top 10 Fourier coefficients by magnitude; we label time points as "spiking" (1) or "not spiking" (0) according to whether their amplitude is greater than the median amplitude in the training data. We then take their corresponding frequencies $f_i$ and represent each time point in $\mathbb{S}^1$ via the

following transformation:

$$\phi : \mathbb{R}^1 \to (\mathbb{S}^1)^{10}, \ \phi(t) = \left( \cos\left( 2\pi \frac{t}{f_i} \right), \ \sin\left( 2\pi \frac{t}{f_i} \right) \right) \Big|_{i=1}^{10} \tag{82}$$

This yields a product space representation in $(\mathbb{S}^1)^{10}$. For each benchmarking trial, we randomly sample 800 points from the training set and 200 points from the test set. We plot both signals, along with their reconstruction using their top 10 Fourier components and the train-test split, in Figure 7.

**Global temperature by month.** We downloaded a list of global average monthly temperatures for the 400 largest cities in the world from Wikipedia (Wikipedia, 2024). We transform longitude and latitude into 3-D coordinates to represent our data in $\mathbb{S}^2$. To convert months to $\mathbb{S}^1$ valued coordinates, we transform ordinal representations of months $t \in [0, 11]$ via the following transformation:

$$\phi : \mathbb{R}^1 \to \mathbb{S}^1, \ \phi(t) = \left( \cos\left( 2\pi \frac{t}{12} \right), \ \sin\left( 2\pi \frac{t}{12} \right) \right) \tag{83}$$

This yields a product space representation of the data in $\mathbb{S}^2 \times \mathbb{S}^1$. For each benchmarking trial, we randomly sample 1,000 (city, month) pairs, and then apply a standard train-test split.

**Traffic prediction.** We download an automobile traffic prediction dataset from Kaggle (Fedesoriano, 2020). This dataset aggregates readings across four sensors with date and time annotations. We process the date and time annotation into day of year ($d$), day of week ($w$), hour ($h$), and minute ($m$) labels and transform to $(\mathbb{S}^1)^4$ analogously to the month timestamps in

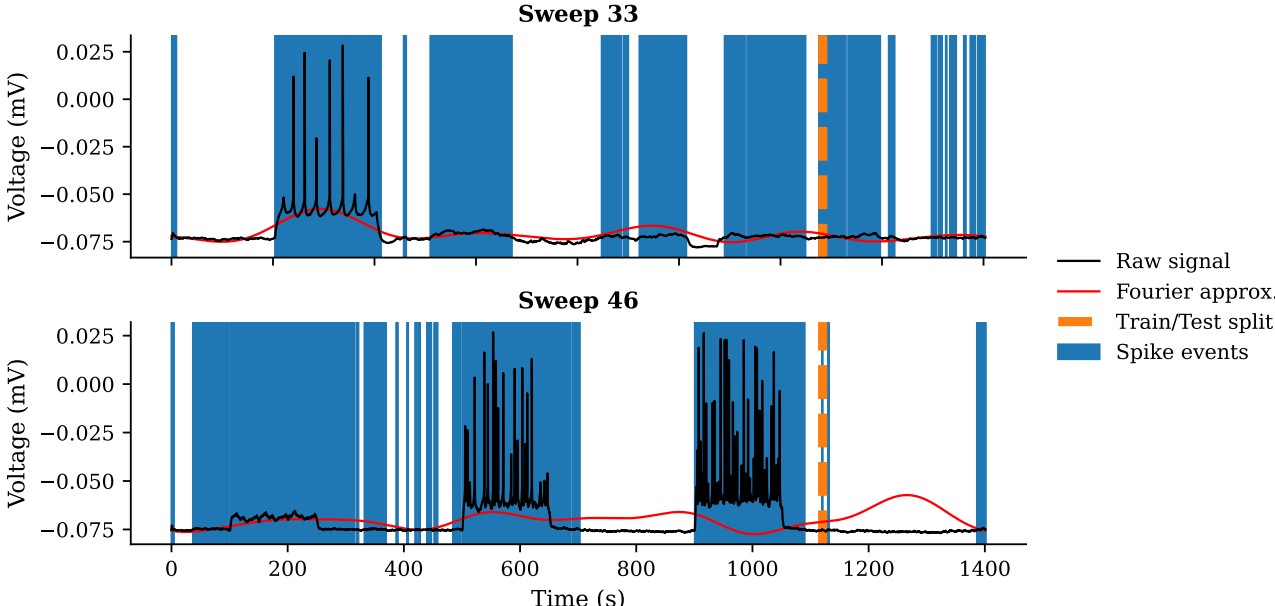

Figure 7: The "Neuron 33" and "Neuron 46" datasets. Their reconstruction using the top 10 Fourier coefficients is shown in red, and the top half of voltages are colored in blue. The first 80% of time points were used to determine Fourier coefficients and model training, while the last 20% are for testing.

the global temperature data. Letting $l$ be the (numeric) label of the sensor, we apply the following transformation to our data:

$$\phi : \mathbb{R}^5 \rightarrow (\mathbb{S}^1)^5 \times \mathbb{E}^1 \tag{84}$$

$$\phi(d, w, h, m, l) = \left( \cos\left(2\pi\frac{d}{365}\right), \ \sin\left(2\pi\frac{d}{365}\right), \right.$$

$$\cos\left(2\pi\frac{w}{7}\right), \ \sin\left(2\pi\frac{w}{7}\right),$$

$$\cos\left(2\pi\frac{h}{24}\right), \ \sin\left(2\pi\frac{h}{24}\right),$$

$$\left. \cos\left(2\pi\frac{m}{60}\right), \ \sin\left(2\pi\frac{m}{60}\right), l \right) \tag{85}$$

## E. Ablations and effects of hyperparameters

For all experiments, we sampled 100 mixtures of 32 Gaussians using the signature $\mathcal{P} = \mathbb{S}^2 \times \mathbb{E}^2 \times \mathbb{H}^2$ in an 8-class regression setting (analogous to the multi-$K$ benchmark in Tables 2 and 3, varying one parameter at a time. Results are plotted in Figure 8.

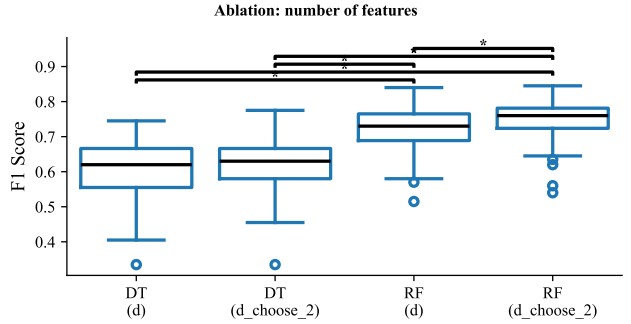

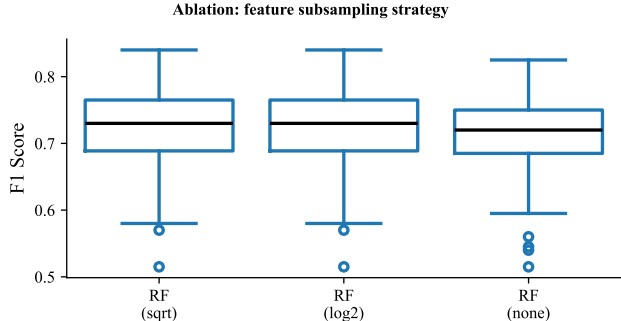

(a) Changing the number of features seen by each DT/RF from 6 to 9 by including the $(x_1, x_2)$ angle is massively beneficial.

(b) Changing feature subsampling approaches in RFs doesn't appear to do much.

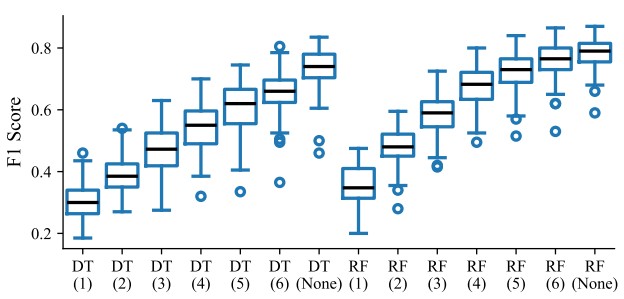

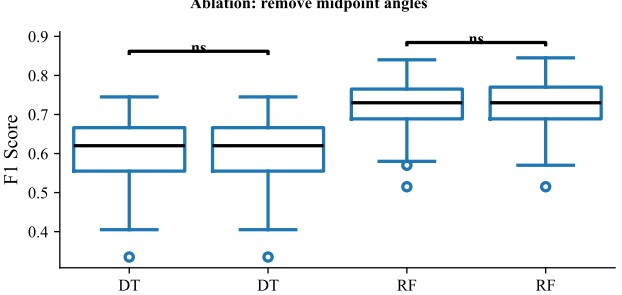

(c) Increasing the maximum depth of each DT/RF is massively beneficial, and shows no signs of overfitting even at unrestricted max depth. All within-predictor differences are significant.

(d) Replacing the midpoint-angle computations with arithmetic means has no statistically significant effect on performance for DTs or RFs, surprisingly.

Figure 8: Effects of various hyperparameters on the performance of our algorithms. Asterisks imply that a result is statistically significant, as determined by the Wilcoxon test with a Bonferroni correction applied; asterisks are omitted for subfigure c, where all changes in depth are significant.

# F. Detailed results

## F.1. Global temperature prediction plots

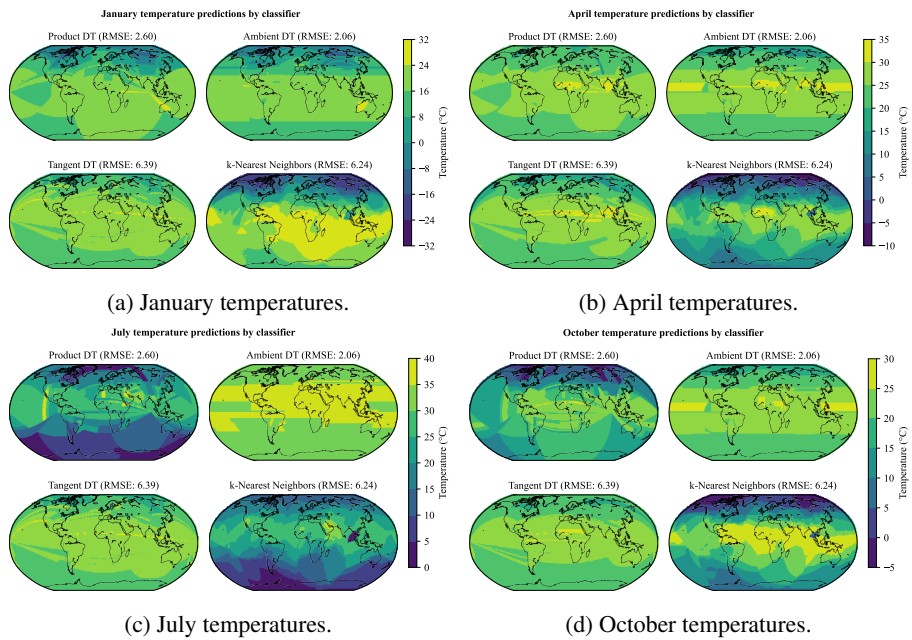

Figure 9: Decision boundaries for the temperature prediction task for the months of January, April, July, and October, colored by predicted temperature across four trained predictors.

## F.2. VAE latent space visualizations

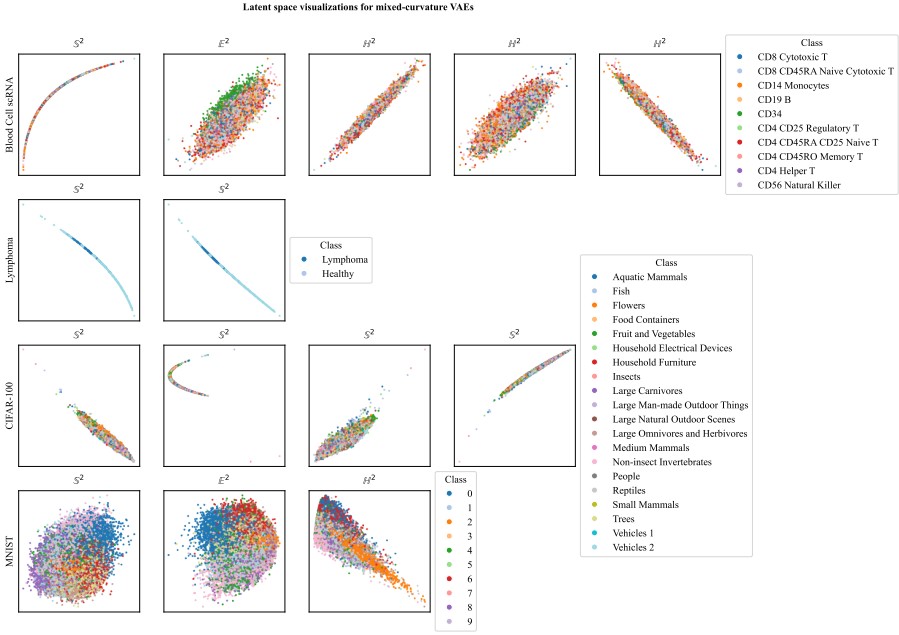

Figure 10: Visualizations of the latent space for all four of the datasets we embed using a VAE, colored by class. For visualization purposes, we show $\mathbb{S}^2$ components in 2-dimensional polar coordinates, and project $\mathbb{H}^2$ embeddings to the Poincaré disk.

# G. Runtimes and complexity

We summarize complexities for models used in this paper, as well as the pairwise distance preprocessing necessary for operations such as computing nearest neighbors and creating reasonable graph edges for GNNs, in Table 7. Complexity estimates are adapted from Virgolin (2021).

To see that the training time complexity of ProductDT is $O(Dnd)$, observe that we must first preprocess the data into angles, which takes $O(nd)$ operations. From there, the angular comparison is a constant-time modification to the decision tree algorithm, so the complexity of ProductDT is $O(nd + Dnd) = O(nd)$. For inference, asymptotic performance is slightly slower than decision trees because preprocessing an input requires $O(d)$ operations.

If using all $\binom{d}{2}$ 2-D projections, training time complexities are all multiplied by $d$, and the $O(d^2)$ preprocessing step is added to test time complexities.

Table 7: Complexity comparison of machine learning models where: $n$: number of samples, $d$: number of features, $h$: neurons per layer, $L$: number of layers, $D$: maximum tree depth, $s$: number of support vectors. We include the complexity of computing pairwise distance, which are necessary for operating models like $k$-nearest neighbors and GNNs without topologies, as well.

| Model | Phase | Time | | Space | |
| --- | --- | --- | --- | --- | --- |
| | | Worst | Avg | Worst | Avg |
| Dists | | $n^2d$ | $n^2d$ | $n^2$ | $n^2$ |
| MLP | Train | $ndh + Lnh^2$ | $ndh + Lnh^2$ | $nd + dh + L(h^2 + nh)$ | $h^2L$ |
| | Test | $h^2L$ | $h^2L$ | $h^2L$ | $h^2L$ |
| Perceptron | Train | $nd$ | $nd$ | $d$ | $d$ |
| | Test | $d$ | $d$ | $d$ | $d$ |
| SVM | Train | $n^3d$ | $n^2d$ | $n^2$ | $n^2$ |
| | Test | $sd$ | $sd$ | $sd$ | $sd$ |
| GNN | Train | $n^2d$ | $n^2d$ | $n^2$ | $n^2$ |
| | Test | $n^2$ | $n^2$ | $n^2$ | $n^2$ |
| k-NN | Train | $1$ | $1$ | $nd$ | $nd$ |
| | Test | $nd + n\log n$ | $\log n$ | $nd$ | $nd$ |
| Decision Tree | Train | $Dnd$ | $Dnd$ | $2^D$ | $2^D$ |
| | Test | $D$ | $D$ | $1$ | $1$ |
| ProductDT (vanilla) | Train | $Dnd$ | $Dnd$ | $2^D$ | $2^D$ |
| | Test | $d + D$ | $d + D$ | $d$ | $d$ |
| ProductDT ($\binom{d}{2}$ splits) | Train | $Dnd^2$ | $Dnd^2$ | $2^D$ | $2^D$ |
| | Test | $d^2 + D$ | $d^2 + D$ | $d^2$ | $d^2$ |

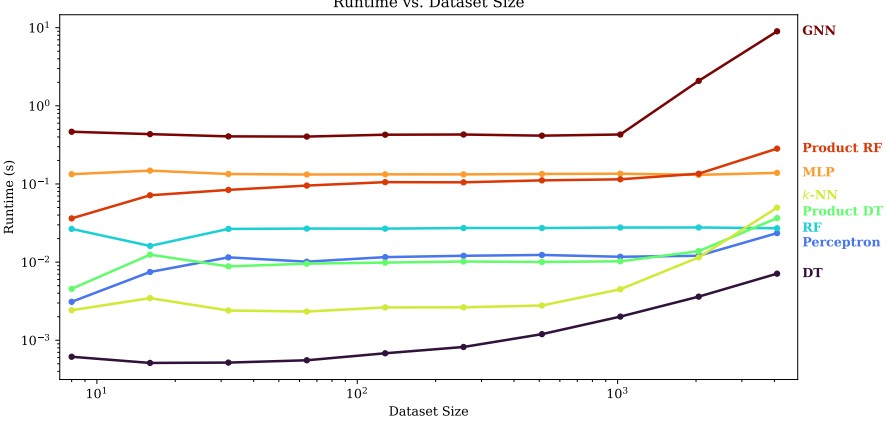

Figure 11: Runtime comparison for all of our methods

# H. Interpretability and visualization

Alongside their demonstrated accuracy and efficiency, decision tree algorithms are attractive for their tractability and interpretability. In particular, given a trained decision tree $\mathcal{T}$, it is possible to:

1. Predict its behavior on the entire space of possible inputs (equivalently: $\mathcal{T}$ partitions $\mathcal{P}$ in a tractable way).

2. Determine the importance of features (for classic decision trees) or feature pairs/components (for ProductDT) by observing how often and how early a feature(/pair/component) is used in the decision tree procedure. Heuristically, early-splitting features are more important.

3. Visualize every node using a 2-dimensional projection of the input data and angle

## H.1. Submanifold-level attribution experiment

To determine whether our method could accurately distinguish between relevant and irrelevant submanifolds, we drew independent samples from Gaussian mixtures in $\mathbb{H}^2$, $\mathbb{E}^2$, and $\mathbb{S}^2$, and yielding datasets $(\mathbf{X}_{\mathbb{H}}, \mathbf{y}_{\mathbb{H}}), (\mathbf{X}_{\mathbb{E}}, \mathbf{y}_{\mathbb{E}}), (\mathbf{X}_{\mathbb{S}}, \mathbf{y}_{\mathbb{S}})$. We then concatenated these embeddings together:

$$\mathbf{X}_{\mathcal{P}} = \mathbf{X}_{\mathbb{H}} \oplus \mathbf{X}_{\mathbb{E}} \oplus \mathbf{X}_{\mathbb{S}}. \tag{86}$$

We trained three separate decision tree models on $\mathbf{X}_{\mathcal{P}}$, using $\mathbf{y}_{\mathbb{H}}, \mathbf{y}_{\mathbb{E}}$, and $\mathbf{y}_{\mathbb{S}}$ as labels. Because the labels and embeddings were drawn independently, it should be the case that only the component from the same manifold as the labels contains any relevant information, and the other two components are simply noise. Therefore, measuring the fraction of splits that fall in the "correct" manifold is a useful proxy for understanding tree models' ability to pick out signal that happens in individual component manifolds.

Our results are summarized in Table 8. We found that both product space and ambient decision trees perform well at this task, which is to be expected.

We note that this analysis is unique to tree methods, where the split dimensions are part of the architecture; other methods, such as perceptrons, $k$-nearest neighbors, or neural networks are harder to query for feature(/component) importances. Therefore, we consider this simple experiment a useful demonstration of how decision tree learning can reveal aspects of structure in mixed-curvature datasets that other learning algorithms cannot reveal.

Table 8: Intepretability outcomes for Gaussian mixture. Percentages reflect the proportion of splits in the trained decision tree which fell in the non-spurious component manifold.

| Model | $\mathbb{H}^2$ | $\mathbb{E}^2$ | $\mathbb{S}^2$ |
|---|---|---|---|
| Product DT | 100% | 83% | 86% |
| Ambient DT | 100% | 83% | 67% |

## H.2. Visualization

A trained tree gives us all of the information we need to visualize the data and how it is split at every node, since each node looks at a 2-dimensional projection. We display three levels of a decision tree with a max depth of 3 in Figure 12. Note that, in this case, the decision tree also gives us relevant information about which 2-dimensional projections are worth looking at on the basis of their feature importances.

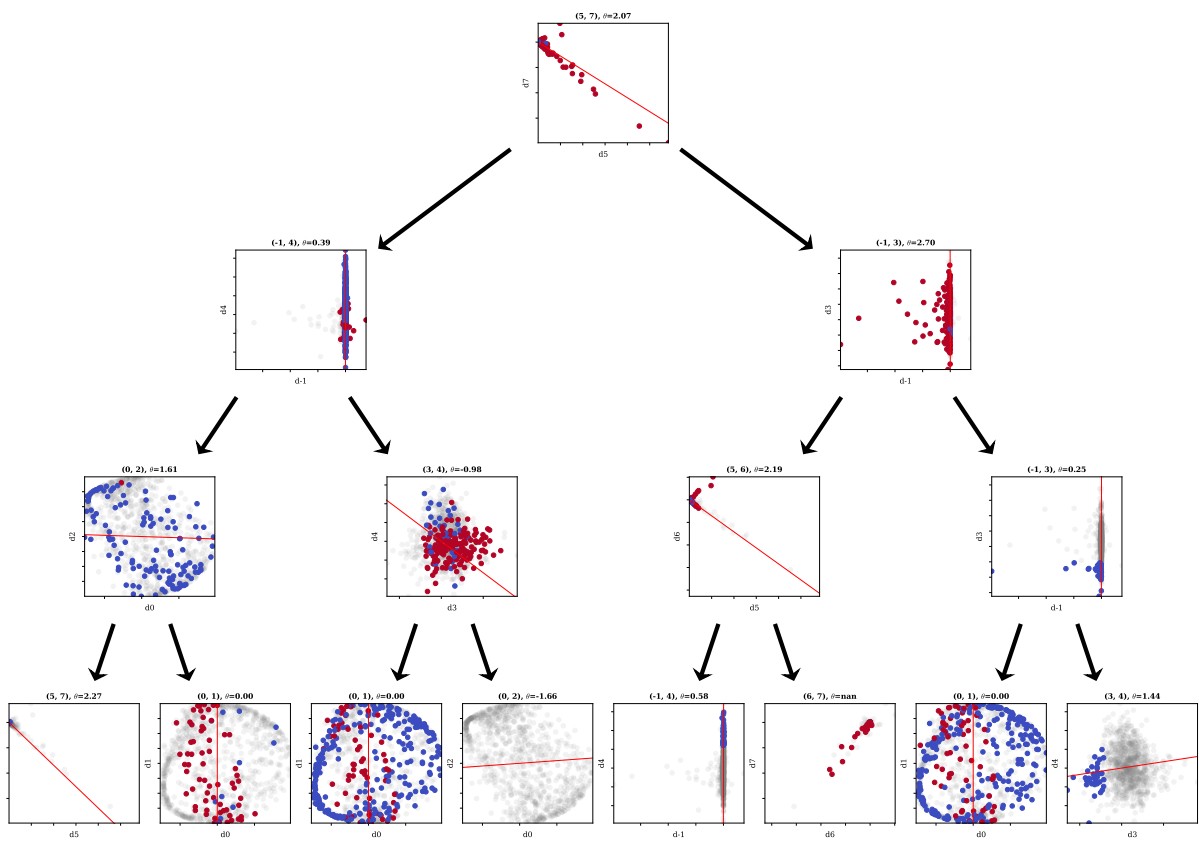

Figure 12: An example of a visualized decision tree for a Gaussian mixture in $\mathcal{P} = \mathbb{S}^2\mathbb{E}^2\mathbb{H}^2$. For each **split**, we show the 2-dimensional projection under which split angles are parameterized. Greyed-out points are discarded earlier in the tree, and therefore do not participate in the split at that level.

# I. Additional results

Table 9: Neural network classification benchmarks.

| | Dataset | Signature | Ambient GCN | Ambient MLP | Ambient MLR | $\kappa$-GCN | $\kappa$-MLP | $\kappa$-MLR | Tangent GCN | Tangent MLP | Tangent MLR |
|---|---|---|---|---|---|---|---|---|---|---|---|
| Synthetic (multi-$K$) | Gaussian | $\mathbb{E}^4$ | 26.5 ± 3.6 | 27.9 ± 3.1 | 28.2 ± 2.8 | 26.5 ± 3.9 | 27.7 ± 2.8 | 27.7 ± 2.7 | 26.5 ± 3.7 | 28.2 ± 2.3 | 28.2 ± 2.9 |
| | | $\mathbb{H}^4$ | 12.0 ± 4.9 | 42.4 ± 5.3 | 40.7 ± 5.6 | 28.8 ± 3.3 | 41.9 ± 4.4 | 43.7 ± 3.7 | 29.8 ± 3.3 | 44.5 ± 3.6 | 44.1 ± 3.5 |
| | | $\mathbb{H}^2\mathbb{E}^2$ | 16.4 ± 4.3 | 29.9 ± 2.7 | 30.0 ± 2.9 | 26.9 ± 2.6 | 28.1 ± 3.2 | 27.2 ± 2.9 | 27.7 ± 3.0 | 31.2 ± 3.7 | 31.3 ± 3.7 |
| | | $(\mathbb{H}^2)^2$ | 12.2 ± 4.2 | 33.7 ± 3.9 | 33.2 ± 4.0 | 26.2 ± 2.4 | 28.7 ± 2.9 | 27.5 ± 3.2 | 28.0 ± 2.1 | 33.5 ± 4.3 | 33.3 ± 4.3 |
| | | $\mathbb{H}^2\mathbb{S}^2$ | 17.6 ± 5.6 | 28.7 ± 2.5 | 29.0 ± 2.4 | 16.4 ± 4.4 | 14.1 ± 3.0 | 15.0 ± 3.6 | 26.2 ± 3.7 | 28.6 ± 3.1 | 28.8 ± 3.2 |
| | | $\mathbb{S}^4$ | 23.9 ± 1.7 | 26.4 ± 1.9 | 25.9 ± 1.6 | 21.1 ± 1.9 | 22.3 ± 1.7 | 21.1 ± 2.0 | 23.5 ± 2.4 | 25.4 ± 2.9 | 25.5 ± 2.8 |
| | | $\mathbb{S}^2\mathbb{E}^2$ | 26.8 ± 2.7 | 23.7 ± 2.0 | 23.9 ± 1.9 | 16.0 ± 2.7 | 17.1 ± 2.9 | 14.1 ± 3.0 | 25.5 ± 2.0 | 25.1 ± 2.3 | 25.2 ± 2.4 |
| | | $(\mathbb{S}^2)^2$ | 23.9 ± 1.7 | 24.3 ± 2.1 | 24.3 ± 2.3 | 15.0 ± 2.2 | 14.1 ± 3.0 | 14.1 ± 3.0 | 24.6 ± 1.8 | 22.8 ± 3.2 | 22.7 ± 3.2 |
| Graph | CiteSeer | $(\mathbb{H}^2)^2$ | 23.8 ± 1.2 | 23.7 ± 1.3 | 23.8 ± 1.3 | 24.9 ± 1.5 | 24.0 ± 1.5 | 25.1 ± 1.3 | 24.8 ± 1.4 | 24.9 ± 1.5 | 24.8 ± 1.5 |
| | Cora | $\mathbb{H}^4$ | 29.5 ± 0.8 | 29.8 ± 0.9 | 29.8 ± 0.9 | 29.6 ± 0.9 | 29.8 ± 0.9 | 29.7 ± 0.9 | 28.9 ± 0.9 | 29.3 ± 0.9 | 29.4 ± 0.9 |
| | PolBlogs | $(\mathbb{S}^2)^2$ | 93.3 ± 1.0 | 92.2 ± 0.9 | 92.1 ± 1.2 | 69.9 ± 10.1 | 70.3 ± 11.4 | 73.4 ± 10.8 | 86.4 ± 4.9 | 84.7 ± 5.1 | 84.9 ± 5.0 |
| VAE | Blood | $\mathbb{S}^2\mathbb{E}^2(\mathbb{H}^2)^3$ | 13.0 ± 1.2 | 12.8 ± 0.9 | 13.0 ± 0.9 | 11.4 ± 0.9 | 11.6 ± 0.8 | 12.2 ± 1.1 | 12.7 ± 1.3 | 12.1 ± 0.6 | 12.1 ± 0.6 |
| | CIFAR-100 | $(\mathbb{H}^2)^4$ | 5.8 ± 0.9 | 7.6 ± 0.9 | 7.5 ± 0.9 | 5.2 ± 0.7 | 5.5 ± 0.9 | 5.3 ± 0.6 | 5.4 ± 0.8 | 7.7 ± 1.0 | 7.7 ± 1.1 |
| | Lymphoma | $(\mathbb{S}^2)^2$ | 78.0 ± 0.4 | 78.0 ± 0.4 | 78.0 ± 0.4 | 66.9 ± 13.9 | 62.8 ± 14.8 | 61.4 ± 15.9 | 78.0 ± 0.4 | 78.1 ± 0.4 | 78.1 ± 0.4 |
| | MNIST | $\mathbb{S}^2\mathbb{E}^2\mathbb{H}^2$ | 18.8 ± 4.4 | 35.9 ± 6.8 | 35.7 ± 6.9 | 14.0 ± 2.9 | 22.5 ± 5.5 | 21.9 ± 5.3 | 18.0 ± 4.3 | 37.4 ± 6.2 | 37.6 ± 6.0 |
| Other | Landmasses | $\mathbb{S}^4$ | 76.1 ± 2.0 | 72.5 ± 2.3 | 71.7 ± 2.1 | 73.2 ± 2.8 | 69.3 ± 3.9 | 70.3 ± 3.5 | 75.1 ± 2.4 | 73.4 ± 1.0 | 73.4 ± 0.9 |
| | Neuron 33 | $(\mathbb{S}^1)^{10}$ | 45.9 ± 2.5 | 45.1 ± 2.0 | 45.1 ± 2.0 | 47.6 ± 2.0 | 46.5 ± 2.5 | 47.0 ± 2.0 | 46.5 ± 2.5 | 55.0 ± 4.0 | 54.8 ± 4.0 |
| | Neuron 46 | $(\mathbb{S}^1)^{10}$ | 50.7 ± 2.0 | 50.6 ± 2.0 | 50.6 ± 2.0 | 49.7 ± 0.8 | 50.4 ± 1.1 | 50.0 ± 0.9 | 51.5 ± 1.2 | 53.7 ± 1.9 | 53.7 ± 1.9 |

Table 10: Neural network regression benchmarks. Missing values represent failed runs.

| | Dataset | Signature | Ambient GCN | Ambient MLP | Ambient MLR | $\kappa$-GCN | $\kappa$-MLP | $\kappa$-MLR | Tangent GCN | Tangent MLP | Tangent MLR |
|---|---|---|---|---|---|---|---|---|---|---|---|
| Synthetic (multi-$K$) | Gaussian | $\mathbb{E}^4$ | 0.21 ± 0.03 | 0.25 ± 0.03 | 0.25 ± 0.03 | 0.21 ± 0.03 | 0.25 ± 0.03 | 0.25 ± 0.03 | 0.21 ± 0.03 | 0.25 ± 0.03 | 0.25 ± 0.03 |
| | | $\mathbb{H}^4$ | 3.12 ± 2.46 | 6.4 ± 9.0E3 | 4.6 ± 4.7E3 | 0.17 ± 0.02 | 0.13 ± 0.02 | 0.13 ± 0.02 | 0.17 ± 0.02 | 0.02 ± 0.00 | 0.02 ± 0.00 |
| | | $\mathbb{H}^2\mathbb{E}^2$ | 2.33 ± 4.00 | 0.71 ± 0.78 | 0.77 ± 0.78 | 0.17 ± 0.03 | 0.22 ± 0.03 | 0.24 ± 0.04 | 0.17 ± 0.03 | 0.03 ± 0.00 | 0.03 ± 0.00 |
| | | $(\mathbb{H}^2)^2$ | 85.94 ± 152.83 | 5.9 ± 1.1E4 | 17.79 ± 31.58 | 0.17 ± 0.03 | 0.21 ± 0.03 | 0.20 ± 0.03 | 0.17 ± 0.03 | 0.02 ± 0.00 | 0.02 ± 0.00 |
| | | $\mathbb{H}^2\mathbb{S}^2$ | 1.96 ± 3.33 | 7.94 ± 13.50 | 0.63 ± 0.63 | 0.23 ± 0.06 | 0.23 ± 0.04 | 0.28 ± 0.06 | 0.17 ± 0.03 | 0.02 ± 0.00 | 0.02 ± 0.00 |
| | | $\mathbb{S}^4$ | 0.17 ± 0.02 | 0.18 ± 0.03 | 0.19 ± 0.03 | 0.25 ± 0.04 | 0.26 ± 0.04 | 0.26 ± 0.04 | 0.18 ± 0.02 | 0.03 ± 0.00 | 0.03 ± 0.00 |
| | | $\mathbb{S}^2\mathbb{E}^2$ | 0.17 ± 0.04 | 0.22 ± 0.04 | 0.22 ± 0.04 | 0.22 ± 0.05 | 0.28 ± 0.06 | 0.26 ± 0.06 | 0.17 ± 0.04 | 0.03 ± 0.00 | 0.03 ± 0.00 |
| | | $(\mathbb{S}^2)^2$ | 0.18 ± 0.04 | 0.21 ± 0.05 | 0.21 ± 0.05 | 0.30 ± 0.06 | 0.18 ± 0.02 | 0.37 ± 0.05 | 0.18 ± 0.04 | 0.03 ± 0.00 | 0.03 ± 0.00 |
| Other | CS PhDs | $\mathbb{H}^4$ | 3.9 ± 2.0E3 | 3.9 ± 2.0E3 | 3.9 ± 2.0E3 | 3.9 ± 2.0E3 | 3.9 ± 2.0E3 | 3.9 ± 2.0E3 | 3.9 ± 2.0E3 | 3.9 ± 2.0E3 | 3.9 ± 2.0E3 |
| | Temperature | $\mathbb{S}^2\mathbb{S}^1$ | 303.60 ± 25.76 | 306.03 ± 20.71 | 305.89 ± 20.08 | | | | 436.05 ± 22.00 | 407.55 ± 24.70 | 407.33 ± 24.59 |
| | Traffic | $\mathbb{E}^2(\mathbb{S}^1)^4$ | 5.17 ± 0.15 | 1.47 ± 0.07 | 1.50 ± 0.06 | | | | 5.21 ± 0.15 | 1.97 ± 0.07 | 1.97 ± 0.07 |

Table 11: Comparison of **Product RF** versus **Ambient RF** on higher-dimensional mixtures of Gaussians, extending the 4-dimensional cases in Table 2. Asterisks indicate statistical significance ($p < .05$).

| Signature | Product RF | Ambient RF | $p$-value |
|---|---|---|---|
| $\mathbb{H}^4$ | **81.73 ± 1.64** | 79.25 ± 1.33 | 0.0078* |
| $\mathbb{S}^4$ | **64.38 ± 1.42** | 61.18 ± 1.13 | 0.0001* |
| $(\mathbb{H}^2)^2$ | **69.27 ± 1.98** | 68.15 ± 1.87 | 0.1191 |
| $(\mathbb{S}^2)^2$ | **41.05 ± 2.27** | 40.45 ± 2.08 | 0.3330 |
| $\mathbb{H}^2\mathbb{S}^2$ | **60.37 ± 1.89** | 60.17 ± 1.66 | 0.7323 |
| $\mathbb{H}^{16}$ | **92.02 ± 1.12** | 90.60 ± 1.08 | 0.0046* |
| $\mathbb{S}^{16}$ | **52.58 ± 1.68** | 46.28 ± 1.55 | 0.0000* |
| $(\mathbb{H}^8)^2$ | **81.45 ± 1.41** | 80.52 ± 1.44 | 0.1469 |
| $(\mathbb{S}^8)^2$ | **66.52 ± 1.53** | 64.90 ± 1.73 | 0.0441* |
| $\mathbb{H}^8\mathbb{S}^8$ | **74.50 ± 1.66** | 73.33 ± 1.45 | 0.0960 |
| $\mathbb{H}^{64}$ | 93.28 ± 0.96 | **94.22 ± 0.80** | 0.0566 |
| $\mathbb{S}^{64}$ | **71.97 ± 1.04** | 68.37 ± 1.37 | 0.0000* |
| $(\mathbb{H}^{32})^2$ | **87.28 ± 1.43** | 87.15 ± 1.43 | 0.9712 |
| $(\mathbb{S}^{32})^2$ | **42.97 ± 1.81** | 38.03 ± 1.78 | 0.0000* |
| $\mathbb{H}^{32}\mathbb{S}^{32}$ | 81.98 ± 1.58 | **82.12 ± 1.72** | 0.8020 |
| $\mathbb{H}^{1024}$ | 94.48 ± 1.14 | **96.00 ± 0.68** | 0.0020* |
| $\mathbb{S}^{1024}$ | **69.65 ± 1.39** | 63.10 ± 1.93 | 0.0000* |
| $(\mathbb{H}^{512})^2$ | 87.85 ± 1.37 | **88.25 ± 1.33** | 0.5809 |
| $(\mathbb{S}^{512})^2$ | **84.50 ± 1.37** | 83.55 ± 1.34 | 0.0949 |
| $\mathbb{H}^{512}\mathbb{S}^{512}$ | **86.43 ± 1.10** | 83.47 ± 1.05 | 0.0002* |

Table 12: Model comparison with decoupled hyperparameters. Because optimal hyperparameters may differ across models, we independently selected best-performing hyperparameters using 5-fold cross validation and evaluated these models on a held-out test set. All values are accuracy when classifying mixtures of Gaussians in $\mathbb{H}^4\mathbb{E}^4\mathbb{S}^4$. In general, we find that models converge on similar hyperparameters, and out model consistently outperforms other random forests.

| | Product RF | Tangent RF | Ambient RF |
|---|---|---|---|
| CV Accuracy | **70.74** | 70.17 | 69.26 |
| Test Accuracy | **71.53** | 70.07 | 69.27 |
| max_depth | 9 | 9 | 9 |
| max_features | sqrt | sqrt | None |
| min_samples_leaf | 8 | 2 | 1 |
| min_samples_split | 8 | 4 | 2 |
| n_estimators | 24 | 24 | 24 |

## J. Datasets availability

Table 13: All of the datasets used in this paper, with download links and citations. CC-BY-SA is short for the Creative Commons Attribution-ShareAlike license. Allen TOU is the Allen Institute terms of use, found at `https://alleninstitute.org/terms-of-use/`.

| Dataset | Link | License | Citation |
|---------|------|---------|----------|
| CiteSeer | Network Repository: CiteSeer | CC-BY-SA | Giles et al. (1998) |
| Cora | Network Repository: CORA | CC-BY-SA | Sen et al. (2008) |
| Polblogs | Network Repository: Polblogs | CC-BY-SA | Adamic & Glance (2005) |
| CS PhDs | Pajek datasets: PhD students in CS | CC-BY-SA | Johnson (1984) |
| Adjnoun | Network Repository: Adjnoun | CC-BY-SA | Newman (2006) |
| Dolphins | Network Repository: Dolphins | CC-BY-SA | Lusseau et al. (2003) |
| Football | Network Repository: Football | CC-BY-SA | Girvan & Newman (2002) |
| Karate Club | Network Repository: Karate | CC-BY-SA | Zachary (1977) |
| Les Mis | Network Repository: Les Mis | CC-BY-SA | Knuth (1993) |
| Polbooks | Network Repository: Polblooks | CC-BY-SA | Krebs (2004) |
| Blood | 10x Genomics: CD8+ Cytotoxic T-cells | CC-BY-SA | Zheng et al. (2017) |
| Blood | CD8+/CD45RA+ Naive Cytotoxic T Cells | CC-BY-SA | Zheng et al. (2017) |
| Blood | 10x Genomics: CD56+ Natural Killer Cells | CC-BY-SA | Zheng et al. (2017) |
| Blood | 10x Genomics: CD4+ Helper T Cells | CC-BY-SA | Zheng et al. (2017) |
| Blood | 10x Genomics: CD4+/CD45RO+ Memory T Cells | CC-BY-SA | Zheng et al. (2017) |
| Blood | 10x Genomics: CD4+/CD45RA+/CD25- Naive T cells | CC-BY-SA | Zheng et al. (2017) |
| Blood | CD4+/CD25+ Regulatory T Cells | CC-BY-SA | Zheng et al. (2017) |
| Blood | 10x Genomics: CD34+ Cells | CC-BY-SA | Zheng et al. (2017) |
| Blood | CD19+ B Cells | CC-BY-SA | Zheng et al. (2017) |
| Blood | 10x Genomics: CD14+ Monocytes | CC-BY-SA | Zheng et al. (2017) |
| Lymphoma | Hodgkin's Lymphoma, Dissociated Tumor: Targeted, Immunology Panel | CC-BY-SA | 10x Genomics (2020a) |
| Lymphoma | PBMCs from a Healthy Donor: Targeted-Compare, Immunology Panel | CC-BY-SA | 10x Genomics (2020b) |
| MNIST | HuggingFace: MNIST | MIT | Lecun et al. (1998) |
| CIFAR-100 | HuggingFace: CIFAR-100 | None | Krizhevsky (2009) |
| Landmasses | Basemap 1.4.1: `is_land` | None | None |
| Neurons | Allen Brain Atlas | Allen TOU | Jones et al. (2009) |
| Temperature | Wikipedia: List of cities by average temperature | CC-BY-SA | Wikipedia (2024) |
| Traffic | Kaggle: Traffic Prediction Dataset | None | Fedesoriano (2020) |

