# OpenReview forum: "Mixed-curvature decision trees and random forests"
_ICML.cc/2025/Conference — ICML 2025 poster_

### Official Review · Reviewer_3Gbp · 2025-03-11

**Overall Recommendation:** 4

**Summary:**

The paper introduces mixed-curvature decision trees (DTs) and random forests (RFs), which can be used to analyse data living on  product manifolds: combinations of Euclidean, hyperspherical and hyperbolic spaces, allowing for heterogeneous curvature. DTs are reformulated using angles to respect the manifold geometry, ensuring that splits are geodesically convex, maximum margin and composable. The methods are evaluated on classification, regression and link-prediction tasks, where they show strong empirical performance. They are also used to analyse graph embeddings and latent spaces from variational autoencoders.

**Update after rebuttal**. The authors clarified that the work is downstream of signature selection. I broadly buy that generalising DTs and RFs to mixed curvature product manifolds might be novel and interesting. I don't think that statistically significant experimental gains across the board are crucial for acceptance, so will stick with my current score. I defer to other reviewers on whether there are problems with the theoretical analysis.

**Claims And Evidence:**

The authors convincingly argue that DTs and RFs can be generalised to all constant-curvature manifolds by representing data and splits as angles in two-dimensional subspaces. They show competitive performance on a wide range of benchmarks, including classification, regression and link prediction. The datasets include synthetic data, graph-embeddings, mixed-curvature VAE latent space, and empirical datasets. Their methods offer a nice interpolation between linear classifiers (ineffective but interpretable) and neural networks (effective but uninterpretable).

**Essential References Not Discussed:**

The references seem reasonable.

**Experimental Designs Or Analyses:**

I think the experimental designs are reasonable; see above.

**Methods And Evaluation Criteria:**

The methods and evaluation criteria make sense; see above.

**Other Comments Or Suggestions:**

I understand that space is limited, but it feels a shame to have Alg 1 in the appendix. I wonder if a shorter version could be included in the main body.

**Other Strengths And Weaknesses:**

**Strengths**:
- Well-written, clear, convincing text
- The core idea – generalising DTs and RFs to mixed curvature product manifolds – seems novel and interesting. Whilst I wouldn’t expect this to ultimately compete with SOTA deep learning, I buy that this might be an effective, interpretable tool in certain simple cases.
- Benchmarking on classification, regression and link prediction tasks on synthetic data, graph embeddings, mixed-curvature VAE latent spaces and empirical data is pretty exhaustive

**Weaknesses**:
- As the authors acknowledge, they view their work as downstream of signature selection and embedding generation. This avoids the fact that good product manifold embeddings might be difficult to find or expensive, and might limit the practicality of their algorithms.

**Questions For Authors:**

- Can you comment more on signature selection? How difficult is this in practice? You acknowledge that ‘product manifolds are not able to represent all patterns in data’, referencing Borde & Kratsios (2023). What types of data patterns or structures are difficult for product manifolds to capture, and how might these limitations be addressed in future work?
- You mention ‘tradeoffs between DTs and RFs and other high-performing methods, especially graph neural networks when topologies are known’. In what specific scenarios do GNNs outperform your method? What characteristics of the data or task contribute to this difference in performance? (I appreciate that this is a big question...)

**Relation To Broader Scientific Literature:**

The paper builds on previous work on machine learning in product manifolds, including Tabaghi et al. (2021, 2024) and Cho et al (2023). This includes linear classifiers, perceptrons, SVMs, PCAs and Transformers. It generalises work by Doorenbos (2023) and Chlenski (2024) on RFs in hyperbolic space, by permitting mixed curvature. Product manifolds have also been used to embed graphs (Wang 2021) and variously in biology.

**Theoretical Claims:**

The paper provides a proof for the theoretical claim that Euclidean DTs using the angular reformulation are equivalent to classical decision trees that perform thresholding in the basis dimensions (see App C). This demonstrates that, while the presentation is unconventional, it is mathematically equivalent to traditional methods. I didn’t check the maths in detail, but this seems reasonable.

---

> ### Author Rebuttal · Authors · 2025-04-01
>
> We thank the reviewer for their careful attention, and are grateful for their favorable assessment of our “**well-written**, clear, convincing text”, our core idea being “**novel and interesting**,” and our method’s potential to be an “**effective, interpretable** tool.”
>
> **Relationship to signature selection and embedding generation.**
> We view signature selection and embedding generation as a complementary avenue of work and have tried to keep our discussion focused on classification and regression with existing embeddings. Nonetheless, as these are all parts of one process, we have tried to model simple but realistic end-to-end product manifold learning pipelines, starting e.g. from pairwise distances (for graph benchmarks) or feature tables.
> Many other papers emphasize the first half of the pipeline—signature selection and embedding generation—and deemphasize approaches to classification and regression. Our work helps complete the picture by expanding and improving ways to use these embeddings once they are generated. We believe this division of focus allows for directed progress in each area and drives the overall field of non-Euclidean machine learning forward. We are optimistic that advances in either of these subproblems can be combined, e.g. by using mixed-curvature decision trees to analyze state-of-the-art product embeddings.
>
> **Algorithm 1 in main body.**
> We agree that it would be better to include Algorithm 1 in the main body of the paper. If accepted, we will use the extra page in the camera-ready to include this.
>
> **Signature selection.**
> We selected signatures as follows:
>
> - For Gaussians on single manifolds, we try a range of curvatures; on product manifolds, we try signatures with curvatures in $\\{-1, 0, 1\\}$
> - For graphs, we grid-search over signatures and use the one with the lowest embedding distortion
> - For VAEs, we follow \[1\]
> - For empirical datasets, the signature is determined by the specific problem.
>
> Selecting a signature is combinatorially difficult because the number of components, and the dimensionality of each component, must be selected. Common heuristics for searching over signatures include:
>
> * The curvature parameter can be learned smoothly in the range $\[-\\infty, \\infty\]$ by using stenographically-projected manifolds, as in mixed-curvature VAE
> * Signatures can be built up component by component according to a greedy algorithm, as in \[4\]
> * Heuristic-guided Bayesian optimization \[3\] is a more principled and sophisticated approach to this
> * In general, embedding is reasonably fast; training VAEs is slower
>
> Regarding our claim that “product manifolds are not able to represent all patterns in data,” the authors of \[1\] make a *theoretical* claim about product manifold limitations, but provide no empirical examples. It is reasonable to speculate that complex graphs with heterogeneous curvature might fall into this category, in which case there is a pragmatic question to ask about fidelity-complexity tradeoffs in the choice of representation geometries.
>
> **Comparison to GCNs.**
> One way of using a graph’s adjacency matrix is to generate embeddings via metric learning, as in \[2\]; another is simply to use GCNs using some set of features. These are not mutually exclusive: in fact, generating metric embeddings is roughly comparable to pretraining a GCN on link prediction. A promising future direction of work could involve using our decision trees as lightweight, easily trainable, and modestly interpretable probes on representations learned by pretraining GCNs.
>
> In general, our benchmarks did not generally favor GCNs even in situations where a known graph adjacency matrix was introduced; however, we appreciate that GCNs are in principle capable of substantially more complicated workflows than the ones that we benchmarked. We include the comment about tradeoffs to reflect the geometric deep learning community’s preference for GCNs on complex real-world tasks more than as a reflection of what we see in our own benchmarks: in particular, $\\kappa$-GCNs are never the best-performing method in our own Table 2\.
>
> **References.**
> \[1\] Borde and Kratsios (2023). Neural Snowflakes: Universal Latent Graph Inference via Trainable Latent Geometries.
> \[2\] Gu et al (2019). Learning Mixed-Curvature Representations in Product Spaces.
> \[3\] Borde et al (2023). Neural Latent Geometry Search: Product Manifold Inference via Gromov-Hausdorff-Informed Bayesian Optimization.
> \[4\] Tabaghi et al (2022). Linear Classifiers in Product Space Forms.

---

> > ### Comment · Reviewer_3Gbp · 2025-04-06
> >
> > Thanks for the response. I'll stick with my current score.

---

### Official Review · Reviewer_SX2Z · 2025-03-13

**Overall Recommendation:** 2

**Summary:**

This paper presents a novel extension of Decision Trees (DTs) and Random Forests (RFs) to product manifolds, which are Cartesian products of hyperbolic, hyperspherical, and Euclidean spaces. The authors introduce an angular reformulation of DTs that respects the geometry of product manifolds, resulting in geodesically convex, maximum-margin, and composable splits. The proposed method generalizes existing Euclidean and hyperbolic algorithms and introduces new hyperspherical DT algorithms. The researchers evaluate their approach on various tasks, including classification, regression, and link prediction, using synthetic data, graph embeddings, mixed-curvature variational autoencoder latent spaces, and empirical data. Their product RFs demonstrate strong performance, ranking first in 25 out of 57 benchmarks and placing in the top 2 for 46 out of 57 when compared to seven other classifiers.

## update after rebuttal

Thank you to the authors for their responses. I believe it is essential to incorporate this discussion into the next revision of the paper.

**Claims And Evidence:**

Overall, the claims made in the submission supported by clear and convincing evidence.

**Essential References Not Discussed:**

Please refer to any references I have mentioned in the answers to other questions. Note that some may have already been cited in the submission.

**Experimental Designs Or Analyses:**

I may have overlooked some details. I assume the numbers and charts indicate the mean scores and 95% confidence intervals as stated in lines 271-273. How many runs has the author conducted for each experiment? I see in line 926 that "we also employ the same random seed for all RF models." Is that same set of seeds applied to all experiments to compute the mean accuracy and its confidence interval?

Additionally, the results from Product RF and Ambient RF are not statistically significantly different in most cases.

The performance of DTs and RFs is sensitive to the choice of hyperparameters. It is not clear how the hyperparameters are chosen (section E.2). And as long as the method for choosing hyperparameters for different RFs is the same, it is not necessary to "set all DT and RF hyperparameters identically" (line 912).

**Methods And Evaluation Criteria:**

The proposed method is a tree-based approach for classification and regression tasks on the product space. While the rationale for choosing a tree-based method in product space is not well-motivated, the author does compare its performance with other non-tree-based methods. Various types of data are considered in the experiments, but the embedding space for the data is relatively low.

**Other Comments Or Suggestions:**

In Eq.1, $x'$ as a tangent vector should lie in the space $R^d$.

**Other Strengths And Weaknesses:**

Weakness:

In this paper, the author use product space as representation space, but in real world, it is hard to choose the optimal signiture of the product space. Moreover authors restrict to work on product space with small (1, 2 and 4) dimension component manifold and the dimension of the product space is also relatively small (< 10), which further limit the paper.

**Questions For Authors:**

line 976, "Table 6: A summary of the neural models benchmarked in our work", where is the results from tangent/ambient MLP/GNN?

**Relation To Broader Scientific Literature:**

The proposed algorithms seem to be straightforward extensions of some exsiting work, combining the classification method on the product space [Tabaghi et al. Linear classifiers in product space forms.] with the HyperDT framework [Chlenski et al. Fast hyperboloid decision tree algorithms.].

The author's reformulation of the decision tree in Euclidean space (Section 3.1) seems unnecessary and introduces extra computations.

Overall, the method appears somewhat incremental and does not offer significant novelty.

**Theoretical Claims:**

Sec 3, line 191-192, the claim "we
observe that homogenous hyperplanes are geodesically convex in any constant-curvature manifold;" it is not proved or cited. Moreover, the concept "geodesically convex" is not defined and the importance of  "geodesically convex" is not clear.

---

> ### Author Rebuttal · Authors · 2025-04-01
>
> Thank the reviewer for their attention to our manuscript, and in particular for praising the strong performance of our method, the thoroughness of our benchmarks, and the “clear and convincing evidence” of our claims.
>
> **Motivation for a tree-based method and our contribution**
>
> Tree-based methods are extremely popular for Euclidean machine learning and have recently become popular in hyperbolic spaces as well \[1,2\]. We believe tree-based methods can be promising as lightweight, simple, easy to train, and interpretable probes on top of learned neural representations or standalone classifiers/regressors in their own right. Additionally, we address a gap in the literature by providing a detailed survey of product space-valued datasets and predictors:
> - We describe and benchmark an end-to-end product space ML pipeline
> - We describe many variants of existing models (e.g. kappa-GCN regression) for the first time
>
> We also note \[3\] is a well-regarded paper that generalizes Euclidean/hyperbolic GCNs to all constant curvatures and product manifolds.
>
> **Theoretical justification for splits**
>
> We thank the reviewer for pointing out that our paper’s exposition on geodesic convexity (GC) and why it matters for classifiers is too short. We initially believed that it would suffice to defer to other papers describing linear classifiers in non-Euclidean spaces (e.g. \[3\], \[4\], and especially \[1\], which explicitly argues on behalf of GC). We agree a brief but explicit discussion of the geometry of linear splits and its relationship to GC is needed. If accepted, we will use our extra space to expand on this between Sections 2.1 and 2.2 as follows:
> - $S \\subset \\mathcal{M}$ is **geodesically convex** if any $p, q \\in S$ implies the geodesic $\\gamma\_{p,q} \\subseteq S$. We take this for granted in Euclidean linear classifiers, as any separating hyperplane preserves this property. However, this is not automatically true in non-Euclidean spaces, which could be a source of misgeneralization for models \[1,4\].
> - Building on \[5\] Ch. 3.1, if $\\mathcal{M}$ is partitioned by GC $G$ into $A$ and $B$, then $A$ and $B$ are also GC. By contradiction, if $\\gamma\_{u,v}$ (where $u, v \\in A$) were to cross into $B$, it must take a path $A \\to G \\to B \\to G \\to A$. This implies the existence of $p, q \\in G$ such that $\\gamma\_{p,q}$ enters $B$, implying $\\gamma\_{p,q} \\nsubseteq G$, violating the GC property of $G$.
> - \[6\] defines a linear split in product manifolds as $l\_\\mathbf{w}^\\mathcal{P} \= \\operatorname{sign}(\\langle w\_\\mathbb{E}, x\_\\mathbb{E} + \\alpha\_\\mathbb{S} \\sin^{-1}( \\langle w\_\\mathbb{S}, x\_\\mathbb{S} \\rangle ) + \\alpha\_\\mathbb{H} \\sinh^{-1} ( \\langle w\_\\mathbb{H}, x\_\\mathbb{H} \\rangle\_\\mathcal{L} + b),$ where $w\_\\mathcal{M}$ means the restriction of $w \\in \\mathcal{P}$ to some submanifold $\\mathcal{M}$.
> - Under our angular reformulation, $w$ is sparse in all but two dimensions lying in the same submanifold, with no bias term; thus, the split simplifies to $\\langle w\_\\mathcal{M}, x\_\\mathcal{M} \rangle \= x\_0\\cos(\\theta) \- x\_d\\sin(\\theta),$ where $\\theta$ is our splitting angle, and $d$ is the dimension along which we split. By restricting our attention to two dimensions within a single component manifold at a time, our angular reformulation approach bypasses almost all the complexity of performing geodesic splits in product manifolds.
>
> **Clarifying experimental details**
>
> - We show means with 95% confidence intervals in our tables
> - We use 10 trials for each experiment
> - Seeds are shared across RFs for splitting, but each trial uses a different seed. For instance, in figure 3:
>   - Seeds 0-9 are used for K=-4, 10-19 for K=-2, 20-29 for K=-1, etc
>
> **Low dimensionality**
>
> We benchmark lower-dimensional embeddings, as these are more challenging and rely more heavily on the classifier’s ability to match the geometry of the space. However, we agree that it is worth evaluating higher-dimensional datasets; therefore, consistent with \[3\] we have benchmarked manifolds with 16 total dimensions at [https://postimg.cc/MvR5R8Pp](https://postimg.cc/MvR5R8Pp), which we will add to the Appendix.
>
> **Tangent vector space**
> - Thank you, we have fixed this.
>
> **Missing neural net benchmarks**
>
> Thank you for pointing this out, we have a full suite of benchmarks at [https://postimg.cc/gallery/ZbfzMzR](https://postimg.cc/gallery/ZbfzMzR).
> We will add the full benchmarks to the Appendix in the camera-ready.
>
> **References:**
>
> \[1\] Chlenski et al (2024). Fast Hyperboloid Decision Tree Algorithms.
>
> \[2\] Doorenbos et al (2024). Hyperbolic Random Forests.
>
> \[3\] Bachmann et al (2020). Constant Curvature Graph Convolutional Networks.
>
> \[4\] Cho et al (2018). Large-Margin Classification in Hyperbolic Space.
>
> \[5\] Urdiste (1994). Convex Functions and Optimization Methods on Riemannian Manifolds.
>
> \[6\] Tabaghi et al (2022). Linear Classifiers in Product Space Forms.

---

> > ### Comment · Reviewer_SX2Z · 2025-04-04
> >
> > Thank you for answering some of my questions.
> >
> > I feel many of my questions have not been addressed in the rebuttal:
> >
> > 1. The comparable performance between ambient RF and product RF in most cases.
> > 2. The reformulation of the decision tree in Euclidean space (Section 3.1) seems unnecessary and introduces extra computations.
> > 3. The choice of hyperparameters in the implementation of the proposed product RF and other tree-based methods.
> > 4. Choosing the optimal signature of the product space for real data.
> >
> > Additionally, I still feel that 16 is a relatively low dimension, though this can be acknowledged as a limitation of the proposed method.

---

> > > ### Author Response · Authors · 2025-04-08
> > >
> > > We thank the reviewer for their follow-up questions and the opportunity to address outstanding concerns. In our initial rebuttal, we prioritized theoretical aspects and our main contributions. Our intent was to use the limited rebuttal space effectively (in accordance with ICML guidelines, which state that authors "are not expected to respond to every individual point in the reviews.")
> > >
> > > We sincerely appreciate the opportunity to elaborate on the remaining points and present additional benchmarks to further clarify our methodology and results. We hope our additional responses resolve any remaining concerns, and would be grateful if the reviewer would consider revising their evaluation accordingly.
> > >
> > > **Q1: More detailed comparison to ambient RFs:**
> > > To address concerns about our performance relative to ambient random forests, lack of statistical significance, and low dimensionality, we ran an additional suite of benchmarks in which we:
> > > * Directly compared ambient RFs and product RFs, omitting other models
> > > * Increased the number of trials to 30 per signature
> > > * Benchmarked signatures totaling 4, 16, 64, 256, and 1024 dimensions.
> > > * Benchmarked signatures: single hyperboloid (H), single hypersphere (S), product of two hyperboloids (HH), product of two hyperspheres (SS), and a hyperboloid-hypersphere product (HS)
> > > * Included Wilcoxon p-values on mean accuracies.
> > >
> > > To adjust for dimensionality, we made two further modifications relative to the Gaussian mixture benchmarks in the paper: we divide the variance by total dimensionality $d$ to prevent norms from blowing up when sampling the Gaussian mixture, and we set `n_features=d` for the product space manifolds because testing all $\\binom{d}{2}$ combinations is infeasible at higher dimensionalities.
> > >
> > > Product RFs outperformed ambient RFs in 16/20 tests, with 8 being statistically significant ($p \< .05)$.
> > >
> > > [Expanded synthetic benchmark table](https://postimg.cc/r0XnKkB5)
> > >
> > > **Q2: Extra computations for Euclidean submanifolds.**
> > > We agree that the computations described in the paper for Euclidean submanifolds are unnecessary. However, there are tradeoffs to removing them: our current split function (Eq. 16\) assumes an angular reformulation (Eq. 15), necessitating Eqs. 17–19 as a consequence.
> > >
> > > The alternative is to remove Eqs. 17–19 and reintroduce the conventional split (Eq. 14). We have tried this in the past, but found that:
> > > * It did not improve runtime
> > > * Using two distinct split functions introduced extra overhead and compromised code elegance
> > > * The angular *perspective* (distinct from choice of implementation) is still needed for narrative reasons, to unify our preprocessing across all component manifold curvatures
> > >
> > > We initially selected simple hyperparameters to facilitate direct comparison between split geometries with minimal confounding. Specifically, we set maximum tree depth to $\\log\_2 32 \= 5$ to match the 32 clusters in our Gaussian mixture experiments. We recognize, however, that the thorough hyperparameter investigation suggested in the initial review could strengthen our work.
> > >
> > > To further investigate the role of hyperparameters, we ran two more benchmarks:
> > > * Decoupled hyperparameter sweep (random search, 5-fold CV, H4 x E4 x S4)
> > > * Maximum depth sweep (grid search, same manifolds as Q1 benchmarks)
> > >
> > > We tested the same choice of 30 random hyperparameters for each model under 5-fold CV, recording the best CV score and test set accuracy for each method. Due to time and computing constraints, we elected to use one signature (H4 x E4 x S4), reasoning that a signature that was not evaluated previously in the paper would prevent biasing our evaluation in favor of any particular model.
> > >
> > > Our search space was:
> > > * `n_estimators`: \[3, 6, 12, 24\]
> > > * `min_samples_split`: \[2, 4, 8, 16, 32\]
> > > * `min_samples_leaf`: \[1, 2, 4, 8, 16\]
> > > * `max_depth`: \[1, 3, 5, 7, 9, None\],
> > > * `max_features`: \["sqrt", "log2", None\]
> > >
> > > In general, we find that all methods respond to hyperparameters similarly, except for the single-manifold RF baseline, which underperformed in general; product space random forests were the best-performing overall. We speculated that Product RFs’ preference for higher values of `min_samples_leaf` and `min_samples_split` may suggest that it has an easier time partitioning the manifold. To corroborate this, we swept over `max_depth` values using a different set of manifolds, finding that Product RF performance saturates earlier than other models for some manifolds.
> > >
> > > [Decoupled hyperparameter sweep scores](https://postimg.cc/Y4mDr9sp)
> > >
> > > [Max depth sweep figure](https://postimg.cc/Vd24v1Q4)
> > >
> > > **Q4: Signature choice.**
> > > We refer you to our rebuttal to reviewer **3Gbp**: subheading “Signature selection” details how we selected signatures for our benchmarks; subheading “Relationship to signature selection and embedding generation" lays out our philosophy on signature selection in this paper.

---

### Official Review · Reviewer_4Ga8 · 2025-03-14

**Overall Recommendation:** 4

**Summary:**

The manuscript develops a methodology for creating decision trees and random forests (classifiers or regressors) by assuming the data coordinates can be decomposed into products of hyperbolic, hyperspherical, or Euclidean components. It is shown that each of those spaces belongs to a class of "constant curvature manifolds" and how, in each, the decision boundary can be written in terms of a threshold on an angle (where the different geometry of the manifolds contribute to different angle equations). Then, the product decision tree (and product random forest) can be expressed in terms of the local decision trees. The manuscript demonstrates the method on a range of benchmarks, with extra effort required to establish how many real-world benchmarks can be expressed as sampled from a product manifold.

**Claims And Evidence:**

The claims on constructing decision trees in various constant curvature manifolds are clear and convincing, the the construction of a product decision tree from them looks reasonable.

**Essential References Not Discussed:**

Not that I could see.

**Experimental Designs Or Analyses:**

No.

**Methods And Evaluation Criteria:**

The manuscript's evaluation criteria are highly biased toward problems in which the input coordinates can be formulated as products of constant curvature. Indeed, the authors make an effort to demonstrate how to cast many problems into those terms. Still, this bias makes the evaluation tricky and prevents reviewers from asking, "How does it perform on [reviewers-favorite-benchmark]?".

The results are good, but only marginally, i.e., comparable to competing methods.

**Other Comments Or Suggestions:**

None.

**Other Strengths And Weaknesses:**

Strengths:
 * Presenting a unified view of constant-curvature manifolds and a novel formulation for decision trees in Euclidean, Hyperbolic and Hyperspherical cases.
 * Formulation of product decision trees.
 * Ideas for how to apply the presented theoretical construct to real-world cases

Weaknesses:
 * There is a lack of clarity regarding when the method is applicable and when we can expect results that are competitive with other methods. That is, while the method obviously works well on product spaces, it is not clear when those are useful approximations in real-world problems.

**Questions For Authors:**

* Can you provide a list of types of problems where the method is applicable per your experience? This would improve the ability to apply your method and might steer future research into extending this list.

**Relation To Broader Scientific Literature:**

The background on DT and RF is well presented, and the main contribution is clearly explained.

**Theoretical Claims:**

Looks valid to me.

---

> ### Author Rebuttal · Authors · 2025-04-01
>
> We thank the reviewer for their favorable comments on our manuscript, including our “**clear and convincing**” claims, “**good results**,” and the “**extra effort** required to establish” performance on real-world benchmarks
>
> **Selection of benchmarks.**
> Our work makes the admittedly strong assumption that we are working with product manifold-valued data, and our method relies on having product manifold-valued inputs in hand. Although we have tried to create as broad a set of benchmarks as possible, we acknowledge that this can be a limitation of the product manifold approach to machine learning as a whole.
>
> In our paper, we try to model what end-to-end approaches to machine learning on product manifolds might look like, including examples of how different types of datasets can be converted to product manifold coordinates. We hope that, in addition to supporting the value of mixed-curvature decision trees, our paper can serve as an instructive guide for researchers who are intrigued by product manifolds but aren't sure how these concepts apply to their own datasets. An additional advantage of product manifolds is they recover single-manifold and Euclidean geometry as special cases: thus, all single-manifold problems can trivially be viewed through the product manifold lens.
>
> More generally, \[1\] describes a method based on sectional curvature to determine whether a graph warrants a product space representation; in such cases, we would expect:
>
> * Product-space representations to model the data substantially better (in terms of distortion, classification accuracy, etc) than single-manifold representations, and
> * Our methods to be effective for classification/regression
>
> We always welcome suggestions for any further benchmarks that can be formulated in terms of product manifolds, particularly those from application areas not included in our original manuscript.
>
> **Suggestions for applications.**
> Per the reviewer’s request, we also provide an incomplete list of some situations in which product manifold approaches can be helpful:
>
> * Many non-Euclidean machine learning problems involve multiple manifolds. For instance, \[2\] describes hyperbolic convolutional neural networks, in which each channel is hyperbolic. To combine representations in multiple channels, they propose a variant of vector concatenation that stays on the hyperboloid. An alternative way to view this would be that the channels form a product space of individual hyperbolic representations.
> * Similarly, pairwise (or higher-order) interactions between data points with non-Euclidean representations can be recast in terms of product manifolds. An example of this is our reformulation of link prediction as binary classification on a product manifold of (outbound node, inbound node) pairs.
> * To embed pairwise distances or other pairwise dissimilarities, one may use the coordinate learning method described in \[1\]. This is a popular approach to embedding graphs, where heterogeneous curvature can make single constant-curvature manifolds inadequate. Our graph benchmarks follow this approach.
> * To embed features without known pairwise distances, mixed-curvature VAEs can be useful, as in \[3\].
> * We are particularly excited about biological applications of product manifolds, which have been a popular application area for hyperbolic deep learning \[4\], and where complicated latent structures can give rise to heterogeneous curvature. For instance, we follow \[5\] in using mixed-curvature VAEs to embed single-cell transcriptomics data. We speculate that in such datasets, differentiation trajectories may embed in hyperbolic space, whereas periodic signals (e.g. those pertaining to the cell cycle) embed in hyperspherical space. Empirically, \[5\] finds that single-cell data embeds better into product manifolds than single manifolds, and \[6\] embeds pathway graphs in product manifolds.
>
> **References.**
> \[1\] Gu et al (2019). Learning Mixed-Curvature Representations in Product Spaces.
> \[2\] Bdeir et al (2024). Fully Hyperbolic Convolutional Neural Networks for Computer Vision.
> \[3\] Skopek et al (2020). Mixed-curvature Variational Autoencoders.
> \[4\] Khan et al (2025). Hyperbolic Genome Embeddings.
> \[5\] Tabaghi et al (2022). Linear Classifiers in Product Space Forms.
> \[6\] McNeela et al (2023). Mixed-Curvature Representation Learning for Biological Pathway Graphs.

---

> > ### Comment · Reviewer_4Ga8 · 2025-04-08
> >
> > Thank you for your detailed response. I will keep my favorable review and score.

---

### Official Review · Reviewer_5kKt · 2025-03-14

**Overall Recommendation:** 1

**Summary:**

This paper proposes mixed-curvature decision trees (DTs) and random forests (RFs) for data embedded in product manifolds—combinations of hyperbolic, spherical, and Euclidean spaces. The core algorithm selects the geodesic split from three options (hyperbolic, spherical, Euclidean) with highest information gain.

**Claims And Evidence:**

- Performance claims are seemingly contradicted by results in Table 1, where product RFs underperform compared to single-manifold RFs in some cases.
- No proofs or formal guarantees are provided for the decision boundary derivations or optimization steps to support the theoretical foundation.
- Motivation for product manifolds not sufficiently justified; theoretically-solid alternatives like ensembles of single-manifold trees are not discussed.

**Essential References Not Discussed:**

./.

**Experimental Designs Or Analyses:**

- Product MLP/GCN baselines are confusing and not explained.
- Tangent RFs are unclear in design.
- Comparing to tangent RFs is not explained.
- Competitors are not well explained
- Lack motivation for manifold configurations (e.g., why specific curvatures).

**Methods And Evaluation Criteria:**

- Synthetic datasets are not described in detail.
- F1/accuracy scores are omitted which clearly reduces interpretability of the results.

**Other Comments Or Suggestions:**

- Remove tangential details (e.g., PyTorch `arctan2` footnote).
- Condense Sections 2.1–2.2;
- Add some background on DTs.
- Define a decision boundary and decision rule for product manifolds for DTs.

**Other Strengths And Weaknesses:**

#### Strengths
  - Novel integration of mixed-curvature geometries into DTs/RFs.
  - Potential for applications in hierarchical or graph-structured data.
#### Weaknesses
  - Poor attribution (e.g., Riemannian geometry basics cited via a single source instead of specific papers).
  - Ambiguous algorithm description (e.g., tangent RFs).
  - Writing lacks clarity, particularly in Sections 2.1–2.2.
  - The **benefits over an RF ensemble of single-manifold DTs** (hyperbolic + spherical + Euclidean) is _very unclear_. This RF ensemble approach is much cleaner and supported by reliable theory.

**Questions For Authors:**

- How are curvature parameters selected for synthetic/data-driven manifolds?
- Why compare to tangent RFs? How do they differ from product RFs?

**Relation To Broader Scientific Literature:**

- Builds on mixed-curvature learning but does not cite __specific__ foundational works or sections, but generally refers to a single source for its foundation without specific attribution.

**Theoretical Claims:**

1. The decision boundary is unclear.
2. It is unclear how geodesic splits are computed for product manifolds (e.g., interactions between geometries?).
3. The tangent space definition is presented without context, relevance to the algorithm, and  does not directly lead to a tangent RF.
4. Theoretical guarantees for convergence or optimality are absent.
5. No theoretical justification is given
6. Product-manifold decision boundaries are not defined. The paper relies on a ad-hoc approach instead.

---

> ### Author Rebuttal · Authors · 2025-04-01
>
> We thank the reviewer for their time and for acknowledging our “novel integration of mixed-curvature geometries into DTs/RFs” and its “potential for applications in hierarchical or graph-structured data,” and for praising its effectiveness in single-manifold settings.
>
> **Added benchmarks**
> Per the reviewer’s suggestions, we have re-run our benchmarks:
> - We have added ensembles of single-manifold trees, which generally underperform our method
> - We have included F1-macro scores for classification
> - For transparency, we include all neural models.
>
> Accuracies, F1 scores, and MSEs can be found at [https://postimg.cc/gallery/ZbfzMzR](https://postimg.cc/gallery/ZbfzMzR).
>
> **Clarifying benchmark assumptions**
>
> The reviewer suggests that Table 1 contradicts our performance claims. Table 1 shows that our method does well *both* in problems involving single-curvature and product manifolds; these are different problems, not different methods; therefore, they cannot be contrasted in this way. In particular, our benchmarking setup assumes a manifold is already given. For instance, given a dataset with signature H2 x E2 x S2, our only choices are:
> - Coerce to E8 (Ambient models)
> - Coerce to E6 (Tangent models)
> - Coerce to one of {H2, E2, S2} per tree (single-manifold tree ensemble)
> - Our method
>
> Of the above, ours is *the only method* for training a decision tree *without* coercing to a different geometry.
>
> For details about our selection of signatures, please refer to our discussion with 3Gbp, subheading “Signature selection.”
>
> **Comparison to single-manifold decision trees**
>
> We justify our choice not to use single-manifold trees in our paper:
> > Letting a single DT span all components—as opposed to an ensemble of DTs, each operating in a single component—allows the model to independently allocate its splits across components according to their relevance to the task at hand
>
> Nonetheless, we agree it is better to verify. In our benchmarks, these ensembles tended to underperform other models, including ours. Furthermore, because designing an ensemble of single-manifold decision trees relies on the constant-curvature decision trees we describe in Sections 3.1–3.3, this method is still a novel contribution of this paper.
>
> **Theoretical justification**
> - For an elaboration on geodesic convexity and decision boundaries, please refer to our discussion with SX2z, subheading “Theoretical justification for splits.” In brief, we explain and commit to updating our manuscript with:
>   - A definition of geodesic convexity
>   - A proof sketch for why geodesic splits partition manifolds into geodesically convex subspaces
>   - The equation for geodesic splits in product manifolds
>   - Citations for each component manifold being split by homogeneous hyperplanes
>   - An explicit connection between geodesic splits and our angular reformulation, clarifying that our splits are fully contained within one of the component manifolds, bypassing the complex interactions between component manifold geometries
>   - A more explicit restatement of the relationship between this and the discussion of geodesic convexity in hyperbolic decision trees (HyperDT)
> - Optimization details:
>   - We modify CART, a greedy algorithm without optimality guarantees \[1\]. The overall behavior of the algorithm is left intact. Moreover, constructing optimal DTs is known to be NP-hard \[2\].
> - Tangent DTs/RFs:
>   - We directly use the method used in \[3\]: apply the log-map at the origin and then train Euclidean DTs/RFs on that. We highlight Section 4.3 for a description.
>   - Eqs. 1 and 12 are necessary for extending the log map to product manifolds, enabling the use of tangent DTs/RFs
>
> **Clarifying writing**
>
> - We commit to rewriting the caption for Table 1 to make it clearer that the manifolds represent aspects of the *problem*, not of the *approach* used, to prevent future confusion
> - We will consolidate Section 2 for clarity; more specific editing suggestions are greatly appreciated.
> - Model details: detailed descriptions for Tangent DTs/RFs, as well as neural net benchmarks, can be found in Appendix E; we will expand Appendix E to be even more detailed.
> - We will add the citations from our reply to SX2z.
>
> **Algorithm 1 curvature handling**
>
> Alg. 1 intentionally omits curvature handling because only the sign of the curvature (i.e. manifold type) is needed to compute midpoint angles under our formulation.
>
> **Clarifying Appendix C**
>
> The reviewer claims “Synthetic results show unclear performance trends / mixed performance; no ablation on manifold combinations” in Appendix C. However, Appendix C is actually a proof of equivalence for the Euclidean case. We request the reviewer clarify this statement.
>
> **References:**
>
> \[1\] Hastie (2009). The Elements of Statistical Learning: Data Mining, Inference, and Prediction.
>
> \[2\] Hyafil and Rivest (1976). Constructing Optimal Binary Decision Trees is NP-Complete.
>
> \[3\] Chlenski et al (2024). Fast Hyperboloid Decision Tree Algorithms.

---

### Decision · Program_Chairs · 2025-05-01

**Decision:**

Accept (poster)

**Comment:**

This paper investigates decision trees and random forests across geometries of multiple manifolds: Euclidean, hyperbolic, and hyperspherical. This paper received highly mixed reviews, with two reviewers arguing for acceptance and two for rejection. The positive reviewers find the idea novel, the paper well written, and the results good. The negative reviewers find the benefit of product manifold DT over single-manifold DT unclear and find that the paper provides insufficient theoretical foundations. The AC does not agree with the first concern. The authors provide a great deal of additional numbers in the rebuttal, highlighting the improvements over single-manifold DTs. Ambient DTs seem to remain a strong baseline, but the proposed approach is overall better, only not better in all 20 experiments (16/20, 8 significant). The AC does agree with the second point of the reviewers. The submission lacks insights into why geodesic splits partition manifolds into geodesically convex subspaces, along with the required equations. The authors provide the required insights in the rebuttal however. The point that the Euclidean DTs bring unnecessary steps is valid but deemed small and does unify the manifolds. As such, the AC believes that the major concerns by the negative reviewers are either not a big problem or have been addressed in the rebuttal. Hence the AC recommends weak accept.